# TensorRL-QAS: Reinforcement learning with tensor networks for improved quantum architecture search

**Akash Kundu** [ORCID]
QTF Centre of Excellence
Department of Physics
University of Helsinki, Finland
akash.kundu@helsinki.fi

**Stefano Mangini** [ORCID]
QTF Centre of Excellence
Department of Physics
University of Helsinki, Finland,
Algorithmiq
stefano.mangini@helsinki.fi

## Abstract

Variational quantum algorithms hold the promise to address meaningful quantum problems already on noisy intermediate-scale quantum hardware. In spite of the promise, they face the challenge of designing quantum circuits that both solve the target problem and comply with device limitations. Quantum architecture search (QAS) automates the design process of quantum circuits, with reinforcement learning (RL) emerging as a promising approach. Yet, RL-based QAS methods encounter significant scalability issues, as computational and training costs grow rapidly with the number of qubits, circuit depth, and hardware noise. To address these challenges, we introduce *TensorRL-QAS*, an improved framework that combines tensor network methods with RL for QAS. By warm-starting the QAS with a matrix product state approximation of the target solution, TensorRL-QAS effectively narrows the search space to physically meaningful circuits and accelerates the convergence to the desired solution. Tested on several quantum chemistry problems of up to 12-qubit, TensorRL-QAS achieves up to a 10-fold reduction in CNOT count and circuit depth compared to baseline methods, while maintaining or surpassing chemical accuracy. It reduces classical optimizer function evaluation by up to 100-fold, accelerates training episodes by up to 98%, and can achieve 50% success probability for 10-qubit systems, far exceeding the <1% rates of baseline. Robustness and versatility are demonstrated both in the noiseless and noisy scenarios, where we report a simulation of an 8-qubit system. Furthermore, TensorRL-QAS demonstrates effectiveness on systems on 20-qubit quantum systems, positioning it as a state-of-the-art quantum circuit discovery framework for near-term hardware and beyond.

## 1 Introduction

Despite the promise of quantum computing in recent years, the practical realization of quantum advantage remains elusive, primarily due to the limitations of current noisy intermediate-scale quantum (NISQ) hardware [1]. These devices are characterized by restricted qubit counts, limited connectivity, and significant noise, severely constraining the complexity of quantum circuits that can be reliably executed. As a result, most foundational quantum algorithms are not yet feasible on available quantum hardware [2], underscoring the urgent need for novel algorithmic strategies that are both effective and hardware-friendly.

To bridge this gap, hybrid quantum-classical approaches have emerged as a practical solution. Among these, variational quantum algorithms (VQAs) have become the leading paradigm for NISQ devices [3, 4, 5, 6]. VQAs operate by iteratively optimizing the parameters of a parameterized

39th Conference on Neural Information Processing Systems (NeurIPS 2025).

Figure 1: **Schematic representation of *TensorRL-QAS* algorithm**. Given a Hamiltonian for which we seek the lowest eigenstate. By combining tensor network (TN) with reinforcement learning (RL)-based quantum architecture search (QAS), we find solutions that would not be achievable using either approach alone. Inspired by [12], we obtain an approximate ground state of the Hamiltonian using DMRG [13, 14], then use this result to warm-start the QAS training in RL-framework. We employ Riemannian optimization on the Stiefel manifold to map the MPS obtained from DMRG to a quantum circuit [15].

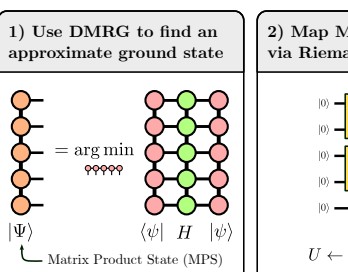
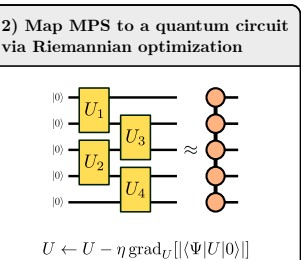
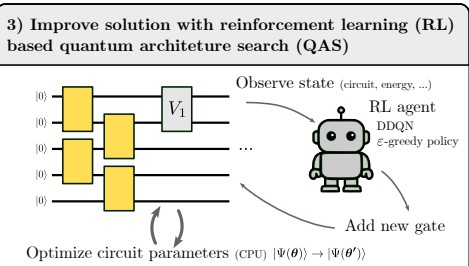

quantum circuit (PQC) to minimize a problem-specific cost function, typically the expectation value of a Hamiltonian $H$. In this framework, a quantum state is prepared using a PQC, $U(\boldsymbol{\theta})$, the cost function $C(\boldsymbol{\theta}) = \langle \mathbf{0} | U^\dagger(\boldsymbol{\theta}) H U(\boldsymbol{\theta}) | \mathbf{0} \rangle$ is measured, and the parameters $\boldsymbol{\theta}$ are updated via a classical optimizer to minimize the cost. The effectiveness of VQAs heavily depends on the choice of the *ansatz* for the underlying PQCs. Typically, PQC architectures are fixed prior to optimization, often guided by hardware connectivity [7] or by problem-specific insights [3, 8, 9, 10]. While these strategies have shown early promise, the resulting circuits are often either too shallow to be expressive or too deep to be noise-resilient, thus limiting the scalability and performance of VQAs [5, 6, 11].

To address these challenges, the field has recently turned its attention to quantum architecture search (QAS) [16, 17, 18], which seeks to automate the discovery of optimal PQC structures from a finite pool of quantum gates. The goal of QAS is to identify an arrangement of gates and corresponding parameters that minimize the target cost function, thereby tailoring circuit architectures to both the problem at hand and the specific constraints of the quantum hardware. Among various QAS strategies, reinforcement learning (RL) has emerged as a particularly promising approach for navigating the vast and complex space of possible circuit architectures [19, 20, 21, 22]. In this framework, the RL-agent sequentially constructs PQCs by selecting gates and their placements, using feedback from the quantum cost function as a reward signal to guide policy updates. Despite the significant potential of RL-based QAS methods, their scalability remains a critical bottleneck. To date, most RL-driven QAS algorithms have only been demonstrated on quantum problems involving up to 8-qubit in noiseless simulations and up to 4-qubit in realistic noisy scenarios. The primary obstacles to scaling include the rapid growth of the action space with increasing qubit number and the need for longer episodes, which together demand a prohibitive number of queries to quantum simulators, let alone real quantum devices. Moreover, the computational cost of simulating quantum circuits grows rapidly with circuit depth, making QAS for larger systems exceedingly challenging (see Appendix L for a detailed discussion).

Our work is motivated by the critical need to overcome scalability limitations in RL-based QAS. To address the dual challenges of action space explosion and prohibitive simulation costs, we introduce **TensorRL-QAS**, a novel framework that combines reinforcement learning with tensor network (TN) methods for QAS. At its core, TensorRL-QAS employs a problem-aware initialization obtained with TN methods, strategically initializing the search space to a physically meaningful starting point [12, 23].

Tensor network methods have become a cornerstone for scalable simulations of quantum systems, providing compact representations of complex, high-dimensional quantum states by means of an efficient representation of entanglement [24, 25]. Their versatility and efficacy have been demonstrated in several tasks, like simulating utility-scale quantum computations [26, 27], characterizing quantum states and operations [28, 29, 30], and for quantum error mitigation [31, 32]. Following the idea originally proposed in [12, 23] on tensor network pre-training of quantum circuits, in this work,

we investigate the integration of TNs and RL by warm-starting an RL-based QAS procedure with TN methods. We schematically summarize the main idea of the manuscript in Fig. 1.

Given a Hamiltonian for which we seek a circuit representation of the ground state, we first use DMRG [13, 14] to find a matrix product state (MPS) approximation of said ground state, then use Riemannian optimization tools to map such MPS to a quantum circuit [15], and eventually refine the circuit using RL-QAS. In particular, we test two different approaches to enhance scaling and reduce the computational cost in RL-QAS: (1) *TensorRL (trainable TN-init)*, in which the TN warm-start is part of the RL-state, and the agent can modify its parameters while training; whereas in (2) *TensorRL (fixed TN-init)*, the TN warm-start is fixed and not part of the RL-state, so the agent cannot modify its parameters. To assess the efficacy of these methods, we compare them against a baseline, where the TN warm-start is replaced with a circuit having the same structure but zeroed-out parameters, namely *StructureRL*. While not a method of independent interest, it enables a direct comparison and demonstrates convergence with the other approaches, given sufficient training.

Through extensive benchmarks on molecular ground-state problems up to 12-qubit, both in noiseless and noisy regimes, TensorRL-QAS consistently produces more compact circuits, while maintaining or surpassing chemical accuracy, and shortens execution times, for example, reducing function evaluations by classical optimizer up to 100-fold and accelerating training episodes by up to 98% compared to baseline QAS methods. By integrating well-established TN methods into RL-based QAS, our methodology enables an improved exploration of PQC architectures, reducing the gap to a practical application of VQAs on NISQ hardware and beyond.

## 2   Related works

Reinforcement learning (RL) has become a prominent framework for optimizing parameterized quantum circuits (PQCs) in variational quantum algorithms, where agents are trained via tailored reward functions to select effective quantum gate sequences. Approaches such as double-deep Q-networks (DDQN) with $\epsilon$-greedy policies have been used for ground state estimation [21], while actor-critic and proximal policy optimization methods have enabled the construction of multi-qubit entangled states [33, 34]. Deep RL has also facilitated hardware-aware circuit design and improved efficiency [20], with recent work incorporating curriculum learning, pruning [35] for quantum architecture search (QAS), and tensor decomposition for $T$-count minimization [36]. In the scope of benchmarking RL-based QAS, ref. [37] introduces a random agent (RA) QAS where the agent chooses the action at each step randomly from a uniform distribution in the RL-framework. Despite these advances, RL-based methods are often constrained to circuits with only a few qubits due to the exponential growth of the action space and the high computational cost per episode.

Alternative non-RL-based strategies include sampling-based methods for circuit selection in ground state estimation [16, 17], Monte Carlo sampling for PQC discovery in QFT and Max-Cut [18], and supernet-based weight sharing for quantum chemistry tasks [38]. Simulated annealing has also been explored for PQC optimization [39]. Meanwhile, evolutionary algorithms such as EQAS-PQC [40] have been introduced to construct PQCs more effectively. Differentiable search techniques such as quantumDARTS [41] leverage Gumbel-Softmax sampling for variational algorithms. In ref. [42], the author introduces a path- and expressivity-based training-free QAS. Neural predictor-based approaches [43] have also been proposed, where a neural network guides quantum architecture search by predicting circuit performance, enabling more efficient exploration of the search space. While these approaches expand the landscape of quantum circuit optimization, scalability and robustness under noise remain open challenges in both noisy and noiseless settings. For a comprehensive review of QAS, we refer the interested readers to [44, 45, 46].

## 3   Methods

In the same spirit of [12], in this work, we use a tensor network (TN) to warm-start the RL-based quantum architecture search (QAS) (see Appendix A for a discussion on how our approach differs from previous TN and VQA synergetic methods). Specifically, we focus our investigation on problems where the goal is to find a parameterized quantum circuit (PQC) that approximately prepares the ground state of a given Hamiltonian of interest. Our approach consists of three main steps, illustrated in Fig. 1. Given a target Hamiltonian $H$, the procedure goes as follows

1. Use density matrix renormalization group (DMRG) [13, 14] with a maximum allowed bond dimension $\chi$ to find a matrix product state (MPS) $|\Psi\rangle$ approximation of the ground state of $H$.

2. Map such MPS to a PQC $|\Psi\rangle \rightarrow U|\mathbf{0}\rangle$ using efficient TN contractions and Riemannian optimization on the Stiefel manifold [15, 47, 48].

3. Continue with RL-based QAS to add more gates to the circuit $U|\mathbf{0}\rangle \rightarrow VU|\mathbf{0}\rangle$ to further minimize the energy and thus better approximate the desired ground state.

Clearly, if DMRG alone accurately obtains the desired ground state, subsequent steps are unnecessary. However, if it fails at maximum bond dimension $\chi$—a parameter that governs both the computational cost and the expressiveness of the underlying tensor network model— the solution can be improved via access to a quantum computer combined with the RL-QAS framework.

## 3.1 DMRG and MPS

Density matrix renormalization group (DMRG) [13, 14, 49, 50] is a well-established and arguably the most used procedure for finding ground states of quantum chemistry Hamiltonians [51]. It does so by solving the optimization problem

$$|\Psi\rangle = \underset{|\psi\rangle, \||\psi\|=1}{\arg\min} \ \langle\psi|H|\psi\rangle \ , \tag{1}$$

where $|\psi\rangle$ is the many-body wavefunction, and the solution $|\Psi\rangle$ is a state modelled by a so-called matrix product state (MPS), which is a classically efficient parameterization of high-dimensional vectors. MPS are also known as tensor trains in numerical linear algebra literature [52, 53]. An MPS for a system of $N$-qubit can be written explicitly as

$$|\Psi\rangle = \sum_{i_1,\ldots,i_N=1} A_{i_1}^{[1]} A_{i_2}^{[2]} \cdots A_{i_n}^{[N]} |i_1 i_2 \ldots i_N\rangle , \quad \begin{cases} A_{i_k}^{[k]} \in \mathbb{C}^{d_k}, & k \in \{1, N\} \\ A_{i_k}^{[k]} \in \mathbb{C}^{d_k} \times \mathbb{C}^{d'_k}, & k \neq \{1, N\} \end{cases} . \tag{2}$$

where $A_{i_1}^{[1]}$ and $A_{i_N}^{[N]}$ are row- and column-vectors, respectively, and the rest are matrices. The shapes are such that the whole vector-matrices-vector multiplication is well-defined. Importantly, all of these have size at most $d_k \leq \chi \in \mathbb{N}$. The hyperparameter $\chi$ is called *bond dimension*, and controls the amount of correlations (entanglement, in the case of quantum) that the MPS can represent. The larger the bond dimension, the more expressive the model is, and thus is capable of representing more general vectors. Briefly, DMRG works by iteratively sweeping through each site in the MPS and optimizing the corresponding local matrix to minimize the cost in (1). This sweeping is continued until convergence, with the computational cost of the whole procedure scaling linearly with the number of sites and polynomially with the bond dimension $\chi$. An in-depth explanation of DMRG and the MPS ansatz is outside the scope of this work, we refer the interested reader to [14, 49, 50].

## 3.2 Mapping MPS to PQC via Riemannian optimization

Given the approximate solution $|\Psi\rangle$ in MPS form, we proceed by finding a quantum circuit that prepares $|\Psi\rangle$. That is, we seek a unitary $U$ such that $U|\mathbf{0}\rangle \approx |\Psi\rangle$. Several proposals have been put forward to address this task, some including explicit constructions [24, 54, 55], others based on iterative variational approaches [56, 57, 58, 59]. In this work, we adopt the second approach due to its hardware efficiency, as it does not require ancillary qubits and densely connected topologies, and for its better empirical performance [56, 57]. Furthermore, as the PQC will be further modified by the following RL-based QAS procedure, an approximate, rather than *exact*, reconstruction suffices. We consider unitaries $U$ which can be decomposed as a product of 2-qubit unitaries, namely $U = \prod_{k=1}^{m} U_k$ with $U_k \in \mathrm{U}(4)$. This is because most quantum computers natively implement 2-qubit interactions, and so the circuit can be mapped to real devices with minimum overhead. The actual form of the unitary $U$ can be chosen freely, but in practice it should satisfy some requirements, like matching the topology of the quantum hardware and reducing the number of operations and depth to minimize the detrimental effect of noisy gates. In this manuscript, we focus on unitaries with a brickwork structure, as illustrated in Fig. 1. This hardware-efficient ansatz is particularly suited for devices with linear connectivity [7], though alternative designs could also be considered.

The circuit can be found by maximizing the overlap between the target state $|\Psi\rangle$ and the variational model, that is maximizing the loss function

$$L(U_1, U_2, \ldots) = |\langle\Psi|U|\mathbf{0}\rangle| = \left|\langle\Psi|\prod_{k=1}^{m} U_k|\mathbf{0}\rangle\right| \qquad (3)$$

whose trainable parameters are the 2-qubit unitaries $\{U_k\}$. In Eq. 3, we show an example of a TN representing the overlap. The minimization process is done by first constructing the overlap TN and then using automatic differentiation to compute the gradients and update all unitaries simultaneously. The contraction of the tensor network and the computation of the gradients remain efficient as long as the PQC is shallow enough so that the $\chi$ in the contraction remains feasible.

Fundamentally, the gradient-based update rule preserves the unitarity of the trainable matrices. This can be obtained by employing Riemannian optimization techniques on the Stiefel manifold [47, 48, 15]. At a high level, the procedure works by modifying the (Euclidean) gradients so that the new updated matrices live on the same manifold as the starting ones. This can be done in several ways, for example, using SVD [56] or, as we instead do, with the so-called Cayley transform [47], which was shown to yield better optimization performances in many quantum-related optimization tasks [15]. Following [15], we have adapted `Adam` [60] to perform Riemannian optimization on the Stiefel manifold and use it to minimize the loss in Eq. (3). We refer to Appendix B for an extended discussion and technical derivations, including the full algorithm for Riemannian optimization.

Appendix H.6 provides a detailed benchmarking of how the number of layers in the brickwork affects training performance when approximating ground-state energies across several molecular systems. Moreover, Appendix H.6 demonstrates that the contribution of the MPS-to-circuit mapping in ground-state preparation and that advanced optimization methods (e.g., reinforcement learning, simulated annealing or evolutionary-based) are required to reliably reach the target ground state while simultaneously reducing quantum gate counts and circuit depth.

## 3.3 Tensor-based reinforcement learning

In contrast to conventional methods, where we start from an empty RL-state [35, 21, 20, 19], TensorRL-QAS starts from the warm-started circuit obtained using the procedures described above. The RL-agent receives such a circuit using the binary encoding scheme introduced in [35, 37]. The agent then sequentially adds gates to this initialized structure until a stopping criterion is met, for example, if the maximum number of actions for that episode is reached, or if a target accuracy is achieved (typically, this is defined as the chemical accuracy for quantum chemistry problems). At each time step, the agent receives a modified RL-state representing the circuit with the newly proposed action. Each action is sampled from an action space containing a pool of gates, in our case set to $\{$RX, RY, RZ, CNOT$\}$. To steer the agent towards the target, we use the same reward function $R$ used in [21, 35] at every time step $t$ of an episode (see Appendix M.1 for more details). With this framework in mind, we introduce two different models, graphically represented in Fig. 2.

**TensorRL (trainable TN-init)**[1] In the first method, we encode both the structure and the parameters of the warm-start circuit in the RL-state through the binary encoding, as illustrated in Fig. 2(a). This enables the agent to receive complete information regarding the MPS, and trains its parameters along with those of the newly added gates in the upcoming steps. In this setting, the size of the encoding is $(D_{\text{MPS}} + D) \times N \times (N + N_{\text{1-qubit}})$, where the first $D_{\text{MPS}} \times N \times (N + N_{\text{1-qubit}})$ encodes the MPS circuit structure of depth $D_{\text{MPS}}$. For each depth, the 2-qubit gates are encoded in the first $(N \times N)$ elements, and the 1-qubit gates are encoded into the remaining $(N_{\text{1-qubit}} \times N)$ elements.

**TensorRL (fixed TN-init)** In the second method, illustrated in Fig. 2(b), we do not feed the warm-start circuit to the RL-state, but rather use it as a fixed initial statevector for the following computations. This means that the binary encoding of the current RL-state does not contain information about the MPS, which is used as a fixed warm-start from the agent. Consequently, this reduces the size of the binary encoding tensor from $(D_{\text{MPS}} + D) \times N \times (N + N_{\text{1-qubit}})$ to $D \times N \times (N + N_{\text{1-qubit}})$. In section Sec. 4 we show that this setting significantly reduces the computational time of each episode, and this happens for the following reasons: (*i*) due to the reduced size of the input tensor and number of steps, the quantum statevector simulation time reduces (see Appendix L); (*ii*) as the RL-state size

---
[1]Throughout the manuscript, the ***TensorRL (trainable TN-init)*** (***TensorRL (fixed TN-init)***) and ***TensorRL (trainable)*** (***TensorRL (fixed)***) are used interchangeably to denote the same method.

Figure 2: (a) **In *TensorRL (trainable TN-init)*, an MPS is transformed into a brickwork circuit structure using Riemannian optimization**. The circuit is encoded into a binary encoding to make it visible to the RL-agent. The RL-agent chooses a gate, and the information corresponding to the gate is encoded into a new binary matrix, highlighted in red. In *StructureRL*, we follow the same steps as above but replace the parameters with zero. (b) ***TensorRL (fixed TN-init)* do not directly encode the MPS structure and its parameters into the RL-state**, but the empty PQC is initialized with the MPS wavefunction.

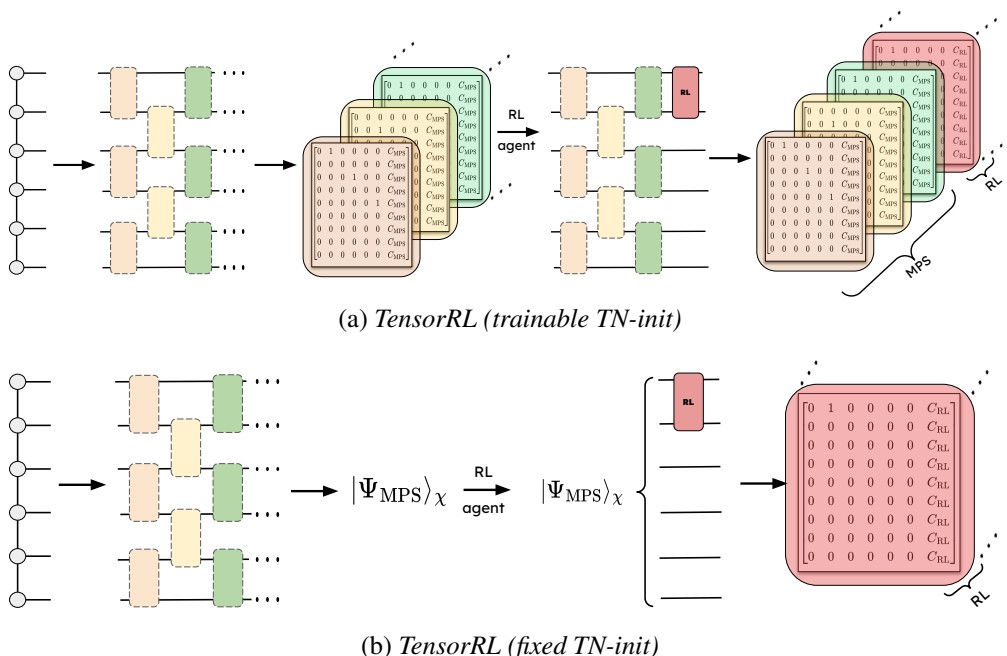

(a) *TensorRL (trainable TN-init)*

(b) *TensorRL (fixed TN-init)*

reduces the neural network receives a reduced input, which further improves the training time per step; and (*iii*) since the warm-start circuit is non-trainable, the number of trainable parameters in the PQC is smaller, which in turn reduces the number of energy evaluations and the time taken by the classical optimizer. A brief discussion of the implementation of TensorRL (fixed TN-init) in quantum hardware is provided in Appendix B.

We additionally compare TensorRL techniques to **StructureRL**, a baseline where the RL-agent is initialized with the warm-start circuit structure (parameters set to zero) and can add gates or modify all parameters, including those in the warm-start section; this isolates the effect of warm-starting versus fixed ansatz initialization. Both *TensorRL (fixed TN-init)* and *TensorRL (trainable TN-init)* yield the most compact PQCs among QAS baselines. With sufficient training, *StructureRL* matches the performance of *TensorRL (trainable TN-init)*, as both share the same initial structure and ultimately optimize similar parameters. Notably, *TensorRL (fixed TN-init)* accelerates episode execution, enabling efficient CPU-only training up to 8-qubit, reducing optimizer calls, and scaling RL-based QAS beyond 12-qubit, including noisy 8-qubit simulations.

## 4 Results

We begin this section by describing the variational quantum optimization problems used to evaluate TensorRL-QAS. We then detail the agent–environment specifications and present the results.

**Experimental details**  We assess TensorRL-QAS on a set of molecular quantum chemistry problems ranging from 6- to 12-qubit, specifically targeting the ground state preparation of 6-$BEH_2$[2], 8-$H_2O$, 10-$H_2O$, and 12-$LiH$ molecules from [61] (see Appendix M.3).  We benchmark against standard RL-QAS baselines, including CRLQAS [35], vanilla RL-QAS [21, 37], RA-QAS [37], training-free

---

[2]Molecules and their corresponding qubit requirements are labelled in the format "$N$-molecule_name", where $N$ indicates the number of qubits.

QAS [42], SA-QAS [39], and quantumDARTS [41]. Appendix E provides detailed descriptions of these methods. Table 1 reports results for two CRLQAS variants, with CRLQAS (rerun) retrained under the same agent–environment settings (discussed below) for fair comparison. Beyond quantum chemistry, Appendix H.5 shows that TensorRL-QAS also outperforms baseline QAS on the 5-qubit Heisenberg and 6-qubit TFIM models, yielding more compact circuits and reduced errors.

**Agent–environment specifications** Our implementation employs the double deep-Q network (DDQN) algorithm [62] with $n = 5$-step trajectory roll-outs [63], discount factor $\gamma = 0.88$, and $\epsilon$-greedy exploration where $\epsilon$ decays geometrically from 1 to 0.05 (decay rate 0.99995/step). The target network is updated every 500 steps, with unified hyperparameters across quantum systems: batch size 1000, replay memory size 20000, `ADAM` optimizer [60] with learning rate $\eta = 3 \times 10^{-4}$, and 5-layer neural networks of 1000 neurons each. All TensorRL-QAS configurations use bond dimension $\chi = 2$ unless otherwise specified, ensuring consistent entanglement structure throughout the search. For DMRG and MPS-to-circuit mapping, we use the tensor network package by `quimb` [64]. The resulting MPS is then mapped to a one-layered brickwork quantum circuit (for example, the first three unitaries in Eq. (3)) using the process explained in Sec. 3.2, and then transpile to {`RX`, `RY`, `RZ`, `CX`}. The maximum number of steps (i.e., gates) per episode is detailed in Appendix M.2, which demonstrates that TensorRL-QAS needs fewer than half the steps per episode compared with baseline RL-based QAS. Moreover, in Appendix. D we elaborately discuss why the DDQN algorithm is utilized instead of other on-policy and off-policy algorithms.

## 4.1 Noiseless simulation

As shown in Tab. 1, *TensorRL (fixed TN-init)* consistently produces more compact circuits while maintaining competitive or better accuracy across all molecular systems. For 6-$BEH_2$, TensorRL (fixed TN-init) achieves $6.8 \times 10^{-5}$ error with only 7 depths, 5 CNOTs, and 12 rotations—an order of magnitude improvement over RA-QAS and SA-QAS with 5-9$\times$ fewer CNOTs. For 8-$H_2O$, it requires merely 6 depth and 9 CNOTs versus 40-100 depth and 69-117 CNOTs for competing methods, while maintaining $8.9 \times 10^{-4}$ error—a 7-13$\times$ efficiency improvement. For 10-$H_2O$, TensorRL (fixed TN-init) achieves $4.1 \times 10^{-4}$ error with just 15 CNOTs. While vanilla RL has slightly lower error, it requires 6$\times$ more CNOTs and 4$\times$ greater depths. For 12-$LiH$, TensorRL (trainable TN-init, $\chi = 2$) achieves the best error using only 37 CNOTs—a 10.5$\times$ reduction versus SA-QAS and 8.7$\times$ versus vanilla RL. The fixed TN-init variant produces an exceptionally compact circuit with 15 depths, 30 CNOTs, and only 9 rotations. Moreover, TensorRL demonstrates perfect consistency, achieving chemical accuracy across 100% of seeds, while vanilla RL succeeds in only 70% of 10-$H_2O$ trials, RA-QAS in 60%, and SA-QAS in none. TensorRL's efficiency advantage scales with system size, growing from 5-9$\times$ CNOT reductions for 6-qubit systems to 10-13$\times$ for 12-qubit systems. Importantly, we note that the numbers reported in the Table for *TensorRL (fixed TN-init)*, only counts the operations added by the agent, which are the only ones it can see, since the warm-start is kept as a statevector and not fed to the RL-state, (see Sec 3.3 for detailed discussion).

To further access the advantage of *TensorRL (fixed TN-init)*, we optimize circuits using a hardware-efficient ansatz (HEA) after the same MPS-to-circuit mapping as a fixed initialization, namely, *TN-VQE*. During the training each parameterized gate is optimized by `COBYLA` (with 1000 iterations). The results are presented in Tab. 1. While TN-VQE produces some improvement, it consistently yields higher errors and less efficient circuits than all RL-based approaches, and often fails to reach chemical accuracy, especially for larger molecules. This demonstrates that even with a high-quality initialization, RL-based circuit construction is crucial for achieving both optimal accuracy and efficiency.

In addition to accuracy and circuit compactness, in Fig. 3 we highlight the distinctive feature of TensorRL, namely its substantial computational efficiency. The left panel shows that *TensorRL (fixed)* consistently requires only $\sim 10^3$ function evaluations per episode, up to 100$\times$ fewer than competing methods, which instead require $10^3$-$10^5$ evaluations. While TensorRL (trainable) initially needs more evaluations, it rapidly converges to similar efficiency within $10^3$ episodes. This reduction in function evaluations translates directly to lower computational overhead, as PQC simulation dominates resource usage. The right panel further underscores TensorRL's practical gains: *TensorRL (trainable)* and *TensorRL (fixed)* reduce per-episode execution time by 80% and 98%, respectively, compared to CRLQAS. These improvements stem from TensorRL's physically motivated initialization and exploration, which avoids expensive evaluation of suboptimal circuits. As a result, TensorRL enables more thorough exploration and optimization within the same computational budget, making

Table 1: **TensorRL finds more compact circuit compared to CRLQAS [35], vanilla RL-QAS [21, 37], random agent (RA) QAS [37], training-free QAS [42], simulated annealing (SA) QAS [39], quantumDARTS [41], and TN-VQE (which consists a fixed tensor network initialization followed by standard hardware efficient ansatz) approaches for 6-**$BEH_2$**, 8-**$H_2O$**, 10-**$H_2O$ and 12-**LiH **molecules**. For TF-QAS [42] the total number of gates includes parameterized XX, YY and ZZ gates, which can further be decomposed into $\{RX, RY, RZ, CX\}$. NA denotes not applicable, implying that the specific data for certain variables are not available.

| Molecule | Method | Error | Depth | CNOT | ROT |
|---|---|---|---|---|---|
| 6-$BEH_2$ | **TensorRL (fixed TN-init)** | $6.8 \times 10^{-5}$ | **7** | **5** | **12** |
| | TensorRL (trainable TN-init) | $6.0 \times 10^{-5}$ | 33 | 22 | 96 |
| | StructureRL | $\mathbf{5.9 \times 10^{-5}}$ | 33 | 21 | 97 |
| | TF-QAS [42] | $1.8 \times 10^{-3}$ | NA | NA | 57 (total gate) |
| | RA-QAS [37] | $5.8 \times 10^{-4}$ | 21 | 26 | 17 |
| | SA-QAS [39] | $5.6 \times 10^{-3}$ | 36 | 45 | 28 |
| | TN-VQE | $5.4 \times 10^{-3}$ | 10 | 8 | 12 |
| 8-$H_2O$ | **TensorRL (fixed TN-init)** | $8.9 \times 10^{-4}$ | **6** | **9** | **15** |
| | TensorRL (trainable TN-init) | $2.0 \times 10^{-4}$ | 36 | 30 | 146 |
| | StructureRL | $\mathbf{1.3 \times 10^{-4}}$ | 33 | 30 | 133 |
| | CRLQAS [35] | $1.8 \times 10^{-4}$ | 75 | 105 | 35 |
| | CRLQAS (rerun) | $1.7 \times 10^{-4}$ | 100 | 85 | 58 |
| | Vanilla RL [21] | $1.7 \times 10^{-4}$ | 96 | 117 | 48 |
| | quantumDARTS [41] | $3.1 \times 10^{-4}$ | 64 | 68 | 151 |
| | RA-QAS | $1.1 \times 10^{-3}$ | 56 | 87 | 41 |
| | SA-QAS | $2.6 \times 10^{-3}$ | 40 | 69 | 26 |
| | TN-VQE | $2.7 \times 10^{-3}$ | 10 | 14 | 6 |
| 10-$H_2O$ | **TensorRL (fixed TN-init)** | $4.1 \times 10^{-4}$ | **17** | **15** | **17** |
| | TensorRL (trainable TN-init, $\chi = 3$) | $6.7 \times 10^{-4}$ | 33 | 34 | 168 |
| | TensorRL (trainable TN-init) | $7.1 \times 10^{-4}$ | 35 | 48 | 173 |
| | StructureRL | $5.8 \times 10^{-4}$ | 37 | 52 | 169 |
| | CRLQAS (rerun) | $3.4 \times 10^{-4}$ | 114 | 140 | 43 |
| | Vanilla RL | $\mathbf{2.5 \times 10^{-4}}$ | 73 | 96 | 32 |
| | RA-QAS | $9.9 \times 10^{-4}$ | 80 | 152 | 58 |
| | SA-QAS | $4.7 \times 10^{-3}$ | 24 | 42 | 14 |
| | TN-VQE | $1.1 \times 10^{-3}$ | 26 | 51 | 14 |
| 12-LiH | **TensorRL (trainable TN-init)** | $\mathbf{1.0 \times 10^{-2}}$ | 31 | 37 | 203 |
| | TensorRL (fixed TN-init) | $2.4 \times 10^{-2}$ | **15** | **30** | **9** |
| | StructureRL | $2.2 \times 10^{-2}$ | 40 | 53 | 179 |
| | CRLQAS (rerun) | $2.4 \times 10^{-2}$ | 32 | 68 | 23 |
| | Vanilla RL | $2.2 \times 10^{-2}$ | 140 | 321 | 94 |
| | RA-QAS | $2.3 \times 10^{-2}$ | 165 | 364 | 86 |
| | SA-QAS | $2.5 \times 10^{-2}$ | 178 | 390 | 88 |
| | TN-VQE | $8.6 \times 10^{-2}$ | 33 | 81 | 29 |

it especially valuable for resource-intensive molecular simulations. Overall, the 10–100× reduction in function evaluations and multiple order-of-magnitude speedup establish TensorRL as both more accurate and markedly more efficient than existing quantum architecture search (QAS) methods.

In Appendix H.2, we show that TensorRL consistently enhances RL-agent trainability, yielding higher accumulated rewards than baselines. As problem size increases, in Appendix H.3, TensorRL maintains a success probability of up to 50%, while other methods rarely exceed 1%. Additional experiments on 8- and 10-qubit $CH_2O$ are reported in Appendix. H, which confirm the advantage of TensorRL-QAS in gate count, circuit depth, and accuracy. Fig. 6 illustrates that TensorRL-QAS converges to chemical accuracy up to 8-qubit on a CPU with better training time than CRLQAS and vanilla RL, enabling practical QAS on a CPU. Finally, in Appendix. H.6 we show that with TensorRL (fixed) we can find a good approximation to the ground energy of the transverse field Ising model up to 20-qubit. These results demonstrate both the versatility and strong performance of

Figure 3: **TensorRL (trainable) and TensorRL (fixed) require 80% and 98% less time, respectively, to execute an episode compared to CRLQAS algorithms**. Meanwhile, TensorRL (trainable) and TensorRL (fixed) improve the number of function evaluations (nfev) by the classical optimizer by 10-100 fold, availing the corridor for more exploration-exploitation by the $\epsilon$-greedy agent in a fixed computational budget. The error bar (on the right figure) is the standard deviation ($\sigma$) from the average over 5 random initializations of the neural network. The molecule in this example is 8-qubit $H_2O$.

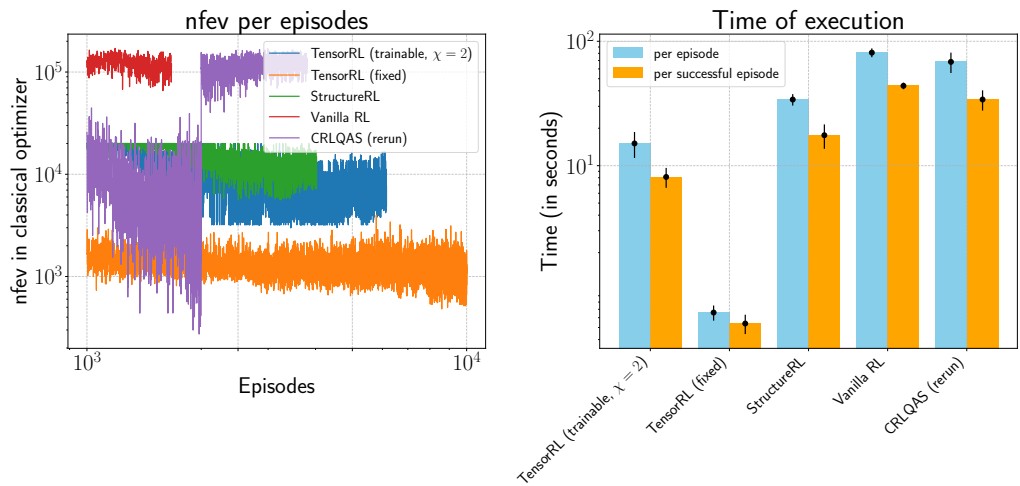

our approach across diverse quantum systems. We further analyze the agent's refinement on tensor network initialization as a function of DMRG bond dimension, with results provided in Appendix Table H.6. This scaling analysis demonstrates that TensorRL offers diminishing improvements as DMRG approaches chemical accuracy, highlighting that the principal advantage of TensorRL-QAS emerges in regimes where classical tensor methods become intractable.

## 4.2 Noisy simulation

In the previous section, we conducted simulations in the ideal noiseless scenario, and now we report results for running TensorRL-QAS in a noisy setting. Specifically, we consider the 8-qubit $H_2O$ molecule ground-state search problem using depolarizing noise rates exceeding current `IBM quantum` [65] hardware: single-qubit gate errors were amplified $10\times$ (i.e. 1%), 2-qubit errors $5\times$ (i.e. 5%), and shot-noise from $10^4$ samples was included (for details see Appendix F). While these conditions do not fully capture all device-specific noise characteristics, they provide a challenging and controlled benchmark for QAS methods. Notably, to the best of our knowledge, this is the largest simulations to date of RL-QAS in a noisy setting, which were previously limited to at most 4-qubit [38, 35]. Under these conditions, TensorRL achieved up to 38% improvement in the error for ground state preparation and $2.4\times$ reduction in average circuit depth compared to the baseline noisy QAS method, CRLQAS [35], indicating improved resilience in both realistic and challenging noise environments. To demonstrate TensorRL-QAS's robustness to hardware noise, we take PQCs proposed during noiseless training that reach the ground state and execute them on `IBMQ-Brisbane` device. Results in Appendix G confirm that TensorRL-QAS can achieve chemical accuracy under realistic hardware noise.

As shown in Table 2, TensorRL (fixed) achieved chemical accuracy with 100% success rate under depolarizing noise, compared to CRLQAS' 30% success rate at comparable error. This performance difference is linked to TensorRL's strategy of preserving noiseless reference states during policy updates, potentially insulating the learning process from noise corruption. The method also produced shallower circuits with half of the CNOT gates compared to CRLQAS, which may help mitigate error accumulation under the amplified 2-qubit noise conditions. Under shot noise, TensorRL maintain an error below chemical accuracy with only a depth 2 circuit, though we note this simplified noise model does not account for correlated errors or crosstalk effects present in physical devices.

Table 2: **TensorRL outperforms baseline CRLQAS under 1- and 2-qubit depolarizing**. We tackle the problem of finding the ground state of $8\text{-}H_2O$.

| Noise | Method | Error | Depth | CNOT | ROT | Success prob. |
|---|---|---|---|---|---|---|
| Depolarizing | **TensorRL (fixed)** | $\mathbf{9.0 \times 10^{-4}}$ | **7** | **5** | **12** | **100%** |
| | TensorRL (trainable) | $2.48 \times 10^{-3}$ | 33 | 22 | 130 | 0% |
| | StructureRL | $2.6 \times 10^{-3}$ | 28 | 22 | 129 | 0% |
| | CRLQAS (rerun) | $1.3 \times 10^{-3}$ | 22 | 11 | 29 | 30% |
| Shot | **TensorRL (fixed)** | $1.5 \times 10^{-4}$ | **3** | **4** | **1** | 100% |
| | TensorRL (trainable) | $\mathbf{8.7 \times 10^{-5}}$ | 30 | 28 | 131 | 100% |
| | StructureRL | $6.8 \times 10^{-4}$ | 34 | 25 | 135 | 100% |

## 5 Conclusion and discussion

We introduced TensorRL-QAS, an improved framework for quantum architecture search (QAS) that combines tensor network initialization with reinforcement learning. Leveraging a matrix product state initialization, TensorRL-QAS reduces function evaluations by $100\times$ and achieves $98\%$ faster per-episode execution compared to baseline QAS methods [42, 35, 18, 66, 39, 37], without sacrificing accuracy. For 6- to-12-qubit chemical Hamiltonians, it consistently yields circuits with $10$–$13\times$ fewer CNOTs and depth, robust performance across random seeds, better efficiency and stability relative to RL and non-RL QAS baselines. The approach generalizes to non-chemical tasks, supports CPU-only training up to 8-qubit, and maintains high accuracy under amplified noise, achieving up to $38\%$ lower energy error and $2.4\times$ smaller circuit depth than existing noisy-QAS methods. These results position TensorRL-QAS as a robust, efficient, and noise-adaptive solution for quantum circuit discovery on near-term devices and in the fault-tolerant quantum computing era [67].

**Limitations and future directions** We study two TensorRL-QAS models: one where the warm-start is part of the RL state (and thus agent-trainable), and one where it is not. Exploring intermediate variants-e.g., where the agent observes but cannot modify the warm-start circuit- remains open. Our action space was limited to single-qubit RX, RY, RZ rotations and CNOT gates; as qubit count and problem complexity grow, richer action spaces may be required. Incorporating problem-specific (e.g., Hamiltonian-inspired) gates [16, 68, 69] or exploiting symmetries and repeated structures (e.g., gadgets [70, 71]) could improve performance beyond 12-qubit systems. Preliminary results with a modified action space presented in Appendix. H.6, moreover, here we also point out the scaling limits of the action space. Further, we only use a brickwork warm-start circuit, matching typical QPU connectivity and MPS structure; future work could examine alternative ansatz and tensor network layouts [72, 59]. Ultimately, due to long queue times on real QPUs, our experiments rely on noise simulations, where we artificially amplify noise by $5$–$10\times$ to test the robustness of TensorRL-QAS. However, we have not conducted training on actual quantum hardware. Future work could focus on real-device training and on developing methods to adapt TensorRL-QAS to actual QPU noise.

In summary, the methods proposed in this work provide state-of-the-art results in quantum architecture search and establish a resource-efficient framework towards scalable reinforcement learning-driven variational quantum optimization. TensorRL-QAS paves the way for practical RL-based quantum circuit design across diverse variational quantum algorithms and larger system sizes. We believe numerous follow-up avenues will reveal the true capability of TensorRL-QAS.

**Code** The source code for all experiments in this manuscript is accessible here: https://github.com/Aqasch/TensorRL-QAS.

## 6 Acknowledgment

The authors acknowledge CSC – IT Center for Science, Finland, for computational resources. AK and SM acknowledge funding from the Research Council of Finland through the Finnish Quantum Flagship Project 358878 (UH). The authors are grateful to the anonymous NeurIPS reviewers and ACs for valuable comments and insights that helped us improve the research.

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

# A   Comparison to prior TN-PQC synergistic frameworks

While both our method and the approach of Rudolph et al. [12] harness tensor networks (TNs) to enhance parametrized quantum circuits (PQCs), there are several key differences between the two works:

- **Integration with reinforcement learning and quantum architecture search:** Rudolph et al.[12] extend the quantum circuit obtained from an MPS warm-start with a fixed PQC structure initialized with random parameters, and then optimize all parameters in the circuit to further decrease the cost function. In contrast, *TensorRL-QAS* does not add a fixed PQC but rather uses reinforcement learning (RL) within a quantum architecture search framework to dynamically explore and optimize the additional PQC architectures that best help in minimizing the loss. Such flexible exploration allows for a balanced search of architectural expressivity and circuit compactness, and hence noise resilience, a capability that is not present in fixed PQC schemes.

- **Trainability of PQCs vs. scalability of RL-QAS:** The main focus of [12] is to mitigate barren plateaus and improve trainability of PQCs through MPS-based initializations. On the contrary, our work focuses on alleviating the heavy computational requirements required by RL-based QAS procedures, thus making them practical also on larger system sizes. Indeed, we show that our approach can achieve *98% reduction in classical computational overhead* by warm-starting RL-framework with MPS warm-starts.

- **Different set of basis gates:** Compared to [12], in this work, we use a different and simpler set of basis gates, which is closer to operations that are natively available on quantum hardware, and thus reduces potential compilation overheads.

- **Different MPS-to-PQC mapping scheme:** In [12], the authors use a combination of explicit constructions and trainable operations to build the PQC approximating the MPS warm-start [57]. In this work, instead, we use a fully variational method to find such a mapping based on Riemannian optimization [15, 59, 73].

# B   Mapping an MPS to a quantum circuit via Riemannian optimization

In this appendix, we explain in more detail the procedure used to find a quantum circuit representation of an MPS using Riemannian optimization.

We consider the case where one fixes the *structure* of the quantum circuit, and aims to find the 2-qubit unitaries that maximize the overlap with the target quantum state. Specifically, let $|\Psi\rangle$ denote the target MPS, and let $U = \prod_{k=1}^{M} U_k$ denote the unitary representing a quantum circuit with a given structure, where each of the $U_k$ is a 2-qubit gate. We then want to solve the following optimization problem

$$\{U_k^*\} = \arg\min_{\{U_k\}} 1 - L(\{U_k\}), \quad L(\{U_k\}) = |\langle\Psi|U|\mathbf{0}\rangle| = \left|\langle\Psi|\prod_{k=1}^{M} U_k|\mathbf{0}\rangle\right| \tag{4}$$

$$\text{with } U_k \in U(4) = \left\{U \in \mathbb{C}^4 \times \mathbb{C}^4 | U^\dagger U = UU^\dagger = \mathbb{I}_4\right\} \forall k = 1, \dots, M,$$

that is, we want to perform optimization on the manifold of unitary matrices, the so-called complex *Stiefel manifold* (and we consider the simpler case of unitary matrices rather than isometries). Such a problem is solved using Riemannian optimization techniques aimed specifically at solving optimization problems on manifolds [74, 48, 15, 47].

We solve the minimization problem above by adapting gradient-descent methods so that each of the trainable parameters of the cost function $L(\cdot)$, namely the matrices $\{U_k\}$, remain unitary during the whole optimization process. Our implementation largely follows [15], and we refer the interested reader to such a reference for an in-depth look at Riemannian optimization, specifically for quantum information tasks. We note that since we are considering optimization over unitary matrices (rather than isometries), the *Euclidean* and *canonical* metric on the Stiefel manifold coincide [74], which greatly simplifies the formulas, for example, compared to those reported in [15]. Additionally, in order to move from one point to another on the manifold, we use *Cayley's retraction*, as it was seen to improve convergence compared to other methods, based for example on SVD [15].

Let $\{U_1^t, \ldots, U_M^t\}$ denote the trainable unitaries at time step $t$ during optimization. Then, following [15, 75], an optimization procedure for adapting `Adam` [60] on the Stiefel manifold can be obtained as follows:

1. Start optimization ($t = 0$) from a point on the manifold, that is, start with unitary matrices $\{U_1^0, \ldots, U_M^0\}$ with $U_k^0 \in U(4) \, \forall k = 1, \ldots, M$. For each of them, set the corresponding initial momentum and velocity terms to zero

$$m_k^{t=0} = v_k^{t=0} = \mathbf{0} \quad \forall k = 1, \ldots, M \, ; \tag{5}$$

where $\mathbf{0}$ is a matrix of shape $4 \times 4$ containing zeros;

2. For $k = 1, \ldots, M$, compute the (Euclidean) gradients of the loss function at the current points

$$\left\{ \frac{\partial L}{\partial U_1^t}, \; \ldots, \; \frac{\partial L}{\partial U_M^t} \right\}, \tag{6}$$

where each of the partial derivatives is itself a $4 \times 4$ complex matrix. Importantly, since the variables are complex, the conjugate of the gradients is used in the following steps $\partial L / \partial U_k^t \to (\partial L / \partial U_k^t)^*$.

3. For $k = 1, \ldots, M$, compute the Riemannian gradients given the (Euclidean) gradients from the previous step (see Eq. (14) in [15])

$$\nabla_R L(U_k^t) = \frac{\partial L}{\partial U_k^t} - U_k^t \left[ \frac{\partial L}{\partial U_k^t} \right]^\dagger U_k^t. \tag{7}$$

4. For $k = 1, \ldots, M$, compute the updated momentum and velocity terms as

$$\begin{aligned}
\tilde{m}_k^{t+1} &= \beta_1 m_k^t + (1 - \beta_1) \nabla_R L(U_k^t), \\
\tilde{v}_k^{t+1} &= \beta_2 v_k^t + (1 - \beta_2) \operatorname{Tr} \left[ \nabla_R L(U_k^t)^\dagger \nabla_R L(U_k^t) \right],
\end{aligned} \tag{8}$$

where $\operatorname{Tr} \left[ \nabla_R L(U)^\dagger \nabla_R L(U) \right] = \langle \nabla_R L(U) | \nabla_R L(U) \rangle$ is the inner product of the Riemannian gradients given the metric. It is the equivalent of the gradient squared term in usual `Adam` (see also [73]).

5. For $k = 1, \ldots, M$, compute the update direction as

$$\tilde{U}_k^{t+1} = m_k^{t+1} \left/ \left( \sqrt{v_k^{t+1}} + \varepsilon \right) \right., \tag{9}$$

where the square root and division are meant element-wise.

6. Apply bias-correction to the learning rate $\eta \to \eta \sqrt{1 - \beta_2^t} / (1 - \beta_1^t)$.

7. For $k = 1, \ldots, M$, move to another point on the manifold using *Cayley's retraction* (see Eq. (15) in [15], specialized for unitary matrices)

$$\begin{aligned}
U_k^{t+1} &= R_{U_k^t}^{\text{Cayley}} \left( -\eta \tilde{U}_k^t \right), \\
R_U^{\text{Cayley}}(V) &:= \left( \mathbb{I}_4 - \frac{W}{2} \right)^{-1} \left( \mathbb{I}_4 + \frac{W}{2} \right) U, \quad W = \frac{V U^\dagger - U^\dagger V}{2}.
\end{aligned} \tag{10}$$

8. For $k = 1, \ldots, M$, use vector transport to the momentum and velocity terms the new points on the manifold (see Eq. (16) in [15])

$$m_k^{t+1} = P_{U_k^{t+1}} \left( \tilde{m}_k^{t+1} \right), \; v_k^{t+1} = P_{U_k^{t+1}} \left( \tilde{v}_k^{t+1} \right) \quad \text{with } P_U(V) := \frac{1}{2} (V - U V^\dagger U). \tag{11}$$

9. Repeat steps (2) to (8) until convergence.

We compute the Euclidean gradients in step (2) using automatic differentiation on a tensor network representation of the loss function in Eq. (4). Alternatively, the gradients with respect to each of the unitaries can be obtained by the standard "punching a hole" method for computing gradients in tensor networks [29, 73]. Also, we remark that we perform a gradient-descent step on *all* unitaries in the variational circuit *at the same time*, that is, we simultaneously update them in a single step of the optimization loop. Other approaches are, however, possible. For example, a DMRG-like optimization procedure could optimize a single unitary while keeping the remaining ones fixed, and then sweep over all unitaries until convergence [59, 56].

## C Implementing *TensorRL (fixed TN-init)* on QPU

Recalling that in the TensorRL-QAS (fixed TN-init) the training begins from a specially prepared tensor network state (such as an MPS). As detailed in Fig. 2(b), rather than embedding information about the initial state in the RL-agent's observations, the MPS is treated as a non-trainable warm-start, significantly reducing the input tensor size and computational requirements.

To translate this idea to real quantum hardware, one can start by directly beginning with the fixed quantum circuit structure corresponding to MPS and then transpile this circuit to the target hardware, ensuring compatibility with device-specific gate sets and connectivity. Any fidelity loss due to hardware noise can be addressed using error mitigation methods such as ref. [31] to better match the realized state to the idealized TN-init. With the TN-init successfully created on the device, the TensorRL algorithm continues from this fixed base state: policies are learned and actions are taken atop this robust initialization, with error mitigation applied as necessary through the computation to maintain performance.

This workflow offers a realistic and accessible route for deploying TensorRL with advanced initialization on hardware. While it necessitates overhead from error mitigation, the resulting process is both experimentally feasible and computationally efficient. Full-scale experimental demonstration is left as future work, given the current constraints in hardware access and resource availability.

## D Why DDQN over other RL-algorithms?

In this section, we will elaborate on the motivation behind utilizing the DDQN algorithm. A direct comparison of different RL algorithms for QAS would be redundant, as the recent BenchRL-QAS [76] study provides a comprehensive benchmark of both policy-based and off-policy agents, including, TPPO [77], PPO [78], A2C [63], A3C [79], DQN [80], Dueling DQN [81], and DDQN [62]. In the setting of ground state preparation with VQE for $6-\text{BEH}_2$, $8-\text{H}_2\text{O}$, and $10-\text{H}_2\text{O}$ molecules, BenchRL-QAS is evaluated under simulated noise, showing that DDQN is the only agent that simultaneously achieves chemical accuracy and the lowest circuit depth. Although these benchmarks are performed in simulated environments, this balance between accuracy and circuit compactness directly translates into practical advantages on real noisy quantum hardware, where reducing gate count lowers error accumulation and increases fidelity. Motivated by these findings, we adopt DDQN as the backbone of TensorRL-QAS, leveraging its suitability for producing hardware-efficient ansatzes while avoiding redundancy with prior benchmarking work.

It is important to note, however, that DDQN may not be the optimal choice across all QAS-related tasks. For example, in Hamiltonian diagonalization or variational quantum classification, policy-gradient methods (e.g., PPO, A2C) and actor–critic architectures often demonstrate better performance. Thus, while DDQN is especially well-suited for VQE-based ansatz construction in noisy-device settings, exploring alternative RL-agents for broader classes of QAS problems remains an interesting direction for future research.

## E QAS algorithms summary

### E.1 CRLQAS: Curriculum reinforcement learning for QAS

The CRLQAS introduced in [35] integrates curriculum-based reinforcement learning with quantum circuit architecture search (QAS) to address the challenges of variational quantum algorithms under hardware noise. In CRLQAS, the agent interacts with an environment where each state is a quantum circuit represented by a tensor-based binary encoding. This 3D binary-encoding captures the gate operations across qubits, circuit depth (moments), and gate types $\{\texttt{CNOT}, \texttt{RX}, \texttt{RY}, \texttt{RZ}\}$, allowing efficient processing by neural networks. The action space is defined by all possible single-qubit rotations and CNOT gates, amounting to $3N + 2\binom{N}{2}$ actions for $N$-qubit. To reduce redundancy and improve learning efficiency, a mechanism for illegal actions is employed, which prevents the agent from appending consecutive identical gates on the same qubit or repeating the same CNOT gate, effectively pruning the action space.

A central feature of CRLQAS is its curriculum learning mechanism, which adapts the difficulty of the optimization task as the agent learns. The curriculum is governed by a moving threshold $\xi$ that determines when an episode is considered successful. This threshold is not static; instead, it is

updated based on the agent's performance using a feedback-driven scheme. Initially, the threshold $\xi_1$ is set empirically. As the agent finds lower energies, the threshold is updated according to

$$\xi_{\text{new}} = |\mu - \xi_2| + \delta,$$

where $\xi_2$ is the best energy found so far, $\delta$ is a small amortization parameter (typically 0.0001), and $\mu$ is the so-called fake minimum energy. The fake minimum energy is calculated as

$$\mu = -\sum_i |c_i|,$$

where $c_i$ are the coefficients in the Pauli decomposition of the molecular Hamiltonian. This value serves as a theoretical lower bound for the ground state energy, ensuring the agent is always guided towards physically meaningful solutions as guaranteed by Rayleigh's variational principle.

The curriculum learning framework further incorporates adaptive rules: after every $G$ episodes (e.g., $G = 500$), the threshold is greedily shifted to $|\mu - \xi_2|$, and after every 50 successful episodes, it is decreased by $\delta/\kappa$ ($\kappa = 10$). If the agent fails to improve the energy for 500 consecutive episodes, the threshold is reset to $\xi_{\text{new}} + \delta$, allowing the agent to escape local minima. This dynamic adjustment ensures that the agent is neither stuck in an overly difficult regime nor allowed to stagnate at suboptimal solutions.

The agent's exploration-exploitation trade-off is managed via an $\epsilon$-greedy policy, where the probability of taking a random action decays exponentially from 1 to 0.05 as training progresses:

$$\epsilon(t) = \max(0.05, 0.99995^t).$$

Experience replay with a buffer size of $20,000$ and double Q-network [62] architectures with periodic target updates are used to stabilize learning. To encourage the discovery of compact circuits, CRLQAS employs a random halting scheme for episodes, where the maximum episode length is sampled from a negative binomial distribution:

$$\Pr(X = n_{\text{fail}}) = \binom{n_{\text{act}} - 1}{n_{\text{fail}}} p^{n_{\text{fail}}} (1 - p)^{n_{\text{act}} - n_{\text{fail}}},$$

with $n_{\text{act}}$ as the maximum number of actions and $p$ the failure probability. This approach increases the likelihood of shorter, more efficient circuits being discovered early in training.

For parameter optimization within each circuit, CRLQAS uses a hybrid Adam-SPSA optimizer. This combines the robustness of simultaneous perturbation stochastic approximation (SPSA) with the adaptive momentum of Adam, and employs a multi-stage shot budget strategy (from $10^3$ to $10^8$ shots) for efficient convergence in noisy environments. The noise simulation leverages the Pauli-transfer matrix (PTM) formalism, with noisy gate operations precomputed offline and accelerated using JAX for GPU, yielding up to a six-fold speedup over conventional CPU-based simulations.

Empirically, CRLQAS achieves chemical accuracy (energy error below $1.6 \times 10^{-3}$ Hartree) for molecules up to 8-qubit, often using fewer gates than baseline methods. The curriculum learning mechanism, driven by the fake minimum energy and adaptive thresholds, enables the agent to autonomously adjust the problem difficulty, maintain exploration, and efficiently find noise-resilient, gate-efficient quantum circuits even under realistic hardware noise profiles. This makes CRLQAS a robust approach for quantum architecture search in both simulated and hardware noisy intermediate-scale quantum settings.

### E.2   TF-QAS: Training-free QAS

TF-QAS [42] is a novel quantum architecture search (QAS) method that aims to eliminate the need for resource-intensive circuit training. It primarily contains the following two steps: (1) *Path-based proxy:* The quantum circuit is represented by a directed acyclic graph [82]. Using the path count between the input-output nodes as a zero-cost complexity metric, this step filters unpromising circuits this enabling rapid pruning of the search space. (2) *Expressibility proxy* Evaluating the remaining circuit using Hilbert space convergence

$$\mathcal{E}(C) = -D_{KL}\left(P_C(F) \,\|\, P_{\text{Haar}}(F)\right), \tag{12}$$

where $P_C(F)$ is the fidelity distribution of random states and $P_{\text{Haar}}$ is Haar-random distribution. Although the progressive strategy balances efficiency in the first step and accuracy in the second, making it practical for NISQ-era quantum devices in Tab. 1, we train TensorRL-agent in finding the ground state of 6-BEH$_2$ and consistently achieves 100-fold better accuracy across all seeds with more quantum quantum circuit, within a smaller time as shown in Tab. 3.

Table 3: **TensorRL converges faster to the optimal circuit structure than TF-QAS**. The times noted for TensorRL are the average time taken by the *TensorRL (trainable TN-init)* and *TensorRL (fixed TN-init)* methods.

| | Method | Training time (h) | Time to solution (h) |
|---|---|---|---|
| $6\text{-BEH}_2$ | TF-QAS | 2.5 | NA |
| | **TensorRL-QAS** | **1.2** | **0.9** |

## E.3 RA-QAS: Random agent for QAS

Following the methodology introduced in [37]. The random agent constructs quantum circuits by selecting actions uniformly at random from the available gate set $\{\texttt{RX}, \texttt{RY}, \texttt{RZ}, \texttt{CNOT}\}$ at each step of an episode. This process is independent of the current circuit state or any reward feedback.

After each randomly selected gate is appended to the circuit, all gate parameters (e.g., rotation angles) are initialized (new gates set to zero, previous gates retain optimized values), and the entire parameter set is optimized using a classical optimizer (COBYLA [83]) to minimize the VQE cost function. This global optimization ensures a fair comparison with learning-based agents, which also rely on classical optimization for parameter updates. Each episode consists of a fixed number of steps $D$, corresponding to the maximum allowed circuit depth. The episode terminates early if the cost function $C_t$ falls below a predefined threshold, i.e the chemical accuracy, indicating successful diagonalization; otherwise, it proceeds until all steps are exhausted.

---

**Algorithm 1** Random Agent for QAS

---

1: **for** episode $= 1$ to $N_{\text{episodes}}$ **do**
2:    Initialize empty quantum circuit $\mathcal{C}$
3:    **for** step $= 1$ to $D$ **do**
4:       Randomly select action $a$ from the gate set $\texttt{RX}$, $\texttt{RY}$, $\texttt{RZ}$
5:       Randomly select target (and control, if $\texttt{CNOT}$) qubits
6:       Append gate $a$ to circuit $\mathcal{C}$
7:       Initialize parameters for new gate(s); retain previous optimized parameters
8:       Optimize all parameters of $\mathcal{C}$ using COBYLA (fixed number of iterations)
9:       Evaluate cost $C_t$ for current circuit
10:      **if** $C_t < \zeta$ **then**
11:         **break**
12:      **end if**
13:    **end for**
14:    Record metrics (success/failure, gate count, circuit depth, absolute error)
15: **end for**

---

As demonstrated in [37], the random agent's performance rapidly degrades with increasing problem size. The number of successful episodes (those that achieve the cost threshold) decreases sharply as the number of qubits grows. Moreover, even in successful cases, the random agent typically produces longer and less efficient circuits compared to reinforcement learning-based approaches. This underscores the necessity of leveraging reward feedback and adaptive strategies for scalable and efficient quantum architecture search.

## E.4 SA-QAS: Simulated annealing for QAS

The algorithm leverages principles from simulated annealing to efficiently explore the space of possible quantum circuit architectures. The algorithm begins with an initial circuit populated with $I$ gates (identity gates). At each iteration $t$, we perform the following steps:

1. Randomly select a position $i$ in the circuit.

2. Replace the gate $\mathbf{A}_i$ with the unitary matrix of a randomly selected quantum gate.

3. Calculate the change in the loss function: $\Delta L_t = L_t - L_{t-1}$.

4. Apply the acceptance criterion:
   - If $\Delta L_t < 0$ (improvement in loss), accept this change.
   - Otherwise, accept this change with probability $e^{-\Delta L_t/T_t}$, where $T_t = \alpha T_{t-1}$.

This process is repeated until either the loss remains stable for a predetermined number of iterations or the maximum iteration limit is reached. The final sequence of gates $\mathbf{A}_0, \ldots, \mathbf{A}_{M-1}$ is submitted as our solution. The algorithm employs a geometric cooling schedule $T_t = \alpha T_{t-1}$. The initial temperature $T_0$ and the cooling rate $\alpha$ are both configurable hyperparameters that control the exploration-exploitation trade-off. Higher initial temperatures encourage more exploration early in the search process, while the cooling rate determines how quickly the algorithm transitions to exploitation. The acceptance probability is defined as

$$P(\text{accept}) = \begin{cases} 1, & \text{if } \Delta L_t < 0 \\ e^{-\Delta L_t/T_t}, & \text{otherwise} \end{cases} \tag{13}$$

This probabilistic acceptance criterion is crucial for escaping local minima in the early stages of the search process, while gradually becoming more selective as the temperature decreases. SA-QAS terminates when either the loss function stabilizes (remains unchanged or changes below a threshold) for a predetermined number of consecutive iterations or the maximum iteration limit $M$ is reached. Upon termination, the sequence of gates $\mathbf{A}_0, \ldots, \mathbf{A}_{M-1}$ represents our discovered quantum circuit architecture.

In our implementation, we maintain a set of candidate quantum gates $\{\text{RX}, \text{RY}, \text{RZ}, \text{CNOT}\}$ from which random replacements are drawn. The loss function $L$ quantifies how well the current circuit configuration performs on the target task. The specific form of the loss function depends on the quantum computational task being addressed. The hyperparameters $T_0$ and $\alpha$ significantly impact the search trajectory and should be tuned based on the specific problem characteristics.

## F  Noise modeling

**Depolarizing noise model**  In our simulations, we model gate errors using the depolarizing noise channel, which provides a simple yet effective description of incoherent noise in quantum circuits [84]. For a single-qubit state $\rho$, the depolarizing channel is defined as

$$\mathcal{E}(\rho) = (1-p)\rho + \frac{p}{2}I, \tag{14}$$

where $p$ is the depolarizing error rate and $I$ is the $2 \times 2$ identity matrix. This channel replaces the input state with the maximally mixed state $I/2$ with probability $p$, while leaving it unchanged with probability $1-p$. For multi-qubit systems, the depolarizing channel generalizes to

$$\mathcal{E}(\rho) = (1-p)\rho + p\frac{I}{2^n}, \tag{15}$$

where $n$ is the number of qubits. In our experiments, we amplify the 1-qubit and 2-qubit gate error rates to $10^{-2}$ and $5 \times 10^{-2}$, respectively, compared to currently available `IBM Quantum` [65] devices to assess the robustness of our algorithm. The depolarizing model is particularly suitable for capturing the average effect of noise in circuits with many gates, as it treats all Pauli errors as equally likely and leads to a simple, analytically tractable form of noise [84].

**Shot noise**  also known as finite-sampling noise, is an intrinsic source of statistical uncertainty in quantum circuit measurements, arising from the quantum nature of measurement and the discrete, probabilistic outcomes inherent to quantum mechanics [85]. In quantum computing, the expectation values of observables are typically estimated by repeatedly executing a quantum circuit and recording the outcomes-a process referred to as collecting "shots". Each shot yields a single outcome sampled from the underlying quantum probability distribution, and the aggregate of many shots provides an empirical estimate of the expectation value. However, due to the finite number of samples, this estimate is subject to statistical fluctuations that scale as $1/\sqrt{N}$, where $N$ is the number of shots. This shot noise persists even in the absence of hardware errors and is governed by the Central Limit Theorem, which ensures that the distribution of the sample mean approaches a normal distribution as the number of shots increases [85]. While increasing the number of shots can reduce the variance of the estimate, practical constraints on quantum hardware often limit the feasible shot count, making shot noise a significant consideration in the design and analysis of quantum algorithms, especially for noisy intermediate-scale quantum devices.

# G  Performance of TensorRL-QAS under real hardware noise

To assess the practical relevance of TensorRL-QAS, we evaluated quantum circuits discovered during RL training for $8 - H_2O$ under realistic noise from `IBMQ-Brisbane` in Qiskit. We report the best episode (in this episode, we get the most compact circuit that achieves the ground state) and two circuits (random 1 and random 2), randomly sampled from the pool of PQCs proposed by the RL-agent. Table 4 displays the noiseless and hardware-noise error rates.

Table 4: **Noiseless and noisy error for selected TensorRL-generated circuits on** $8 - H_2O$; random 1 (2) are PQCs randomly sampled from RL proposals. The best episode corresponds to the PQC with the smallest gate count.

| Method | Noiseless error | `IBMQ-Brisbane` error |
|---|---|---|
| TensorRL (fixed, best episode) | $9.0 \times 10^{-4}$ | $2.8 \times 10^{-3}$ |
| TensorRL (fixed, random 1) | $3.4 \times 10^{-4}$ | $\mathbf{1.5 \times 10^{-3}}$ |
| TensorRL (fixed, random 2) | $1.2 \times 10^{-3}$ | $3.3 \times 10^{-3}$ |

These results demonstrate the effectiveness of TensorRL-QAS under realistic hardware noise and connectivity constraints. The inclusion of random PQCs sampled from the RL proposal pool further highlights the robustness of the architecture search, showing that typical agent-generated circuits maintain competitive performance. We remark that in general *minimizing the noiseless error does not necessarily result in minimizing the noisy error and vice versa* [35].

# H  Robustness of TensorRL

Table 5: **TensorRL demonstrates robust performance by achieving ground state errors comparable to baseline RL, and other non-RL methods for QAS, while constructing quantum circuits with fewer gates and shallower depth**. Results are averaged over 5 random seeds for finding the ground state of $8\text{-}CH_2O$ and $10\text{-}CH_2O$. For $10\text{-}CH_2O$, the limited gate set `RX, RY, RZ, CX` prevents reaching the true ground state, indicating the need for more expressive gates (such as in ADAPT-VQE [16]).

| Molecule | Method | Error | Depth | CNOT | ROT | Perform |
|---|---|---|---|---|---|---|
| $8\text{-}CH_2O$ | TensorRL (trainable TN-init, $\chi = 2$) | $8.0 \times 10^{-5}$ | 30 | 29 | 130 | 1 |
| | TensorRL (zero angle) | $7.6 \times 10^{-5}$ | 34 | 28 | 132 | 1 |
| | **TensorRL (fixed TN-init)** | $3.2 \times 10^{-5}$ | **16** | **13** | **22** | 1 |
| | CRLQAS (rerun) | $\mathbf{9.1 \times 10^{-6}}$ | 43 | 60 | 31 | 1 |
| | Vanilla RL | $1.1 \times 10^{-5}$ | 56 | 44 | 61 | 0.9 |
| | RA-QAS | $1.3 \times 10^{-3}$ | 64 | 107 | 53 | 0.8 |
| | SA-QAS | $7.9 \times 10^{-3}$ | 35 | 59 | 28 | 0 |
| $10\text{-}CH_2O$ | **TensorRL (fixed TN-init)** | $4.5 \times 10^{-3}$ | **15** | **26** | **37** | 0 |
| | TensorRL (trainable TN-init, $\chi = 3$) | $4.3 \times 10^{-3}$ | 39 | 43 | 163 | 0 |
| | TensorRL (trainable TN-init, $\chi = 2$) | $6.1 \times 10^{-3}$ | 34 | 41 | 165 | 0 |
| | StructureRL | $4.7 \times 10^{-3}$ | 34 | 36 | 169 | 0 |
| | CRLQAS (rerun) | $\mathbf{4.2 \times 10^{-3}}$ | 34 | 43 | **23** | 0 |
| | Vanilla RL | $4.3 \times 10^{-3}$ | 344 | 193 | 157 | 0 |
| | RA-QAS | $6.7 \times 10^{-3}$ | 179 | 354 | 120 | 0 |
| | SA-QAS | $8.0 \times 10^{-3}$ | 38 | 70 | **23** | 0 |

## H.1  More molecules

Table 5 provides a comparative analysis of TensorRL against both RL-based and non-RL-based quantum circuit ansatz discovery methods for finding ground states of $8\text{-}CH_2O$ and $10\text{-}CH_2O$ molecules. See Appendix. M.3 for the molecular configuration details. The results demonstrate that TensorRL consistently achieves errors on the same order of magnitude as strong baselines (e.g., CRLQAS,

Vanilla RL), while often constructing quantum circuits with significantly fewer gates and reduced depth.

For 8-$CH_2O$, the *TensorRL (fixed TN-init)* variant achieves a notably low error ($3.2 \times 10^{-5}$) with the shallowest circuit (depth 16) and the fewest CNOT and rotation gates among all tested methods, highlighting its efficiency. For 10-$CH_2O$, although none of the methods reach the true ground state due to a restricted gate set, *TensorRL (fixed TN-init)* still constructs the shallowest circuit (depth 15) with a competitive error ($4.5 \times 10^{-3}$), outperforming other methods in circuit compactness. The experiments also reveal that methods relying solely on the RX, RY, RZ, CX gate set are limited in expressivity for larger molecules, suggesting the need for more expressive gates (as in ADAPT-VQE [16]) to achieve ground state accuracy for challenging systems.

In summary, Table 5 highlights the robustness and efficiency of TensorRL, particularly its *fixed TN-init* variant, in constructing compact quantum circuits while maintaining competitive accuracy compared to both RL and non-RL baselines. These results underscore TensorRL's potential for resource-efficient quantum circuit discovery in quantum chemistry applications.

### H.2 Reward signal analysis

Figure 4 presents the cumulative reward as a function of training episodes for all evaluated methods: TensorRL (trainable, $\chi = 2$), TensorRL (fixed), StructureRL, Vanilla RL [21], and CRLQAS [35] (rerun). The $x$-axis is shown on a logarithmic scale to highlight both early and late training dynamics.

Figure 4: **Cumulative reward vs. episodes for all evaluated methods in the task of finding the ground state of 8-$H_2O$.** All methods are trained for $\sim 48$ hours, and the shaded regions represent the standard deviation across runs. *TensorRL (trainable $\chi = 2$)* shows a sharp increase in accumulated reward compared to other methods, showing enhanced trainability. The shaded area in the plot corresponds to the standard deviation ($\sigma$) from the average over 5 random initializations of the neural network.

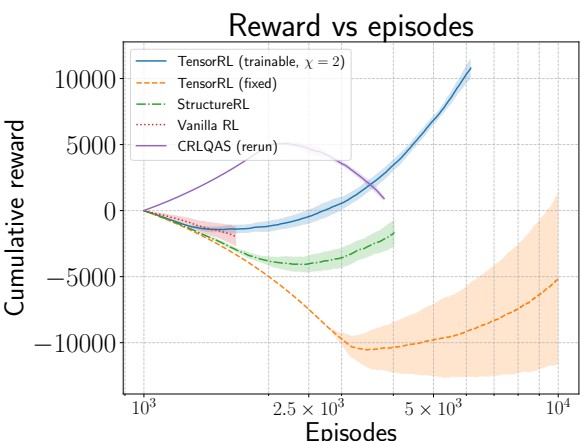

- **TensorRL (trainable, $\chi = 2$):** This method demonstrates the strongest performance, with cumulative rewards increasing rapidly after approximately $2.5 \times 10^3$ episodes. By the end of training, it achieves the highest cumulative reward, exceeding $10,000$, and exhibits low variance, indicating stable and effective learning.

- **TensorRL (fixed):** While this variant outperforms StructureRL and Vanilla RL, its cumulative reward plateaus at a lower value compared to the trainable version. This suggests that trainable parameters in TensorRL are critical for achieving superior performance.

- **StructureRL:** This method shows moderate improvement, with cumulative rewards remaining positive but not reaching the levels of either TensorRL variant. The variance is also relatively contained, suggesting consistent, albeit limited, learning progress.

- **Vanilla RL:** The baseline method struggles to achieve significant positive rewards, with its curve remaining near zero throughout training. This highlights the difficulty of the task and the necessity of structural enhancements.
- **CRLQAS (rerun):** This method initially suffers a sharp decline in cumulative reward, reaching values below $-10,000$ around $5 \times 10^3$ episodes. While some recovery is observed, the method remains substantially below the others in final performance and exhibits high variance.

Overall, these results demonstrate that the proposed TensorRL (trainable) approach significantly outperforms baseline and alternative methods, both in terms of final cumulative reward and learning stability. The advantage of trainable structure within the RL framework is particularly evident, as fixed or less-structured baselines lag in both reward magnitude and convergence speed.

### H.3 Success rate with scaling

Here we investigate Fig. 5, where we plot the probability of success of finding a parameterized quantum circuit that finds the ground state of 8- and 10-$H_2O$ molecules with an increasing number of episodes. The results are summarized in the following:

Figure 5: **The probability of finding a successful episode i.e. a quantum circuit that finds the ground state of 8-, 10-$H_2O$ consistently increases with TensorRL (fixed)**. The error bar represents maximum and minimum values across 5 random neural network initializations, highlighting the robustness of TensorRL (fixed) even at its worst initialization compared to other methods.

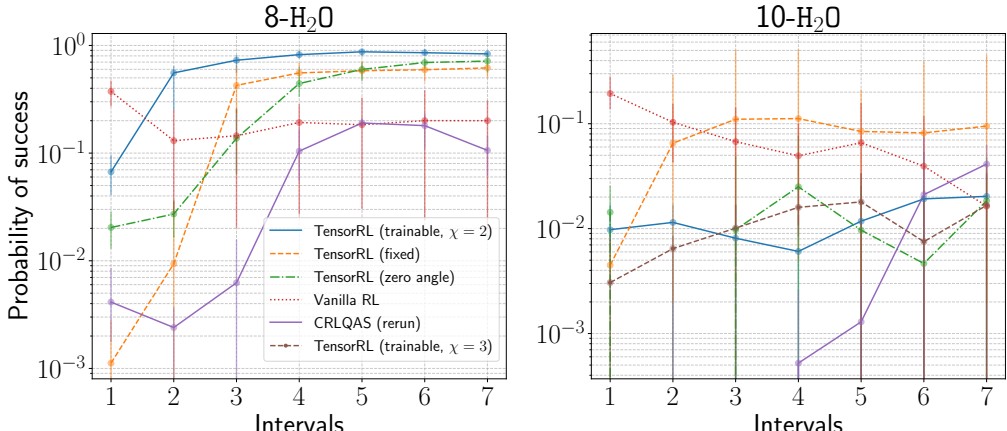

- **TensorRL (fixed) performance** For 8-$H_2O$: Starts with low success ($\sim 0.1\%$) but steadily improves to approximately 50%. The performance curves shown represent maximum and minimum values over 5 random neural network initializations, demonstrating robust training stability. For 10-$H_2O$: While the average success probability remains around $\sim 10\%$, the maximum reaches an impressive 50% success rate for some initializations. This significantly outperforms other methods, which struggle to reach even 1% success at this qubit scale. Exhibits only a $\sim 5{:}1$ performance reduction when scaling up from 8- to 10-qubit. Preserves reasonable success rates at larger scales, making it valuable for practical applications.
- **CRLQAS comparison** For 8-$H_2O$: Begins with extremely low success ($\sim 0.3\%$) and improves to $\sim 10\text{-}20\%$, with visible variability across different initializations. For 10-$H_2O$: Reaches only $\sim 5\%$ at its maximum performance across initializations, despite showing an upward trajectory. Even at its best initialization, CRLQAS fails to approach the 50% maximum success rate achieved by TensorRL (fixed). Shows better scaling properties than trainable models but still substantially underperforms TensorRL (fixed) across all initialization. Demonstrates potential for continued improvement with additional training, but lags significantly behind TensorRL (fixed) in maximum achievable performance.

- **Vanilla RL performance** For 8-$H_2O$: Starts with moderate performance ($\sim$30%) but consistently declines across all initialization. For 10-$H_2O$: Shows continuing performance degradation over time with success rates below 5% for all initializations, never approaching the 50% maximum achieved by TensorRL (fixed). Ultimately falls below TensorRL (fixed) for both system sizes at all initialization configurations. Highlights the limitations of non-tensor-based approaches for quantum circuit optimization, with even its best initialization performing poorly compared to TensorRL (fixed).

TensorRL (fixed)'s average $\sim$10% success probability for 10-qubit systems (with maxima reaching 50% across initializations) means up to half of optimization episodes can yield useful circuits in optimal configurations. This substantially reduces computational overhead for practical quantum simulations. No other method demonstrates this level of success at the 10-qubit scale, with most failing to exceed 1% success rate even at their best initialization. The constrained parameter space of fixed models avoids overfitting while maintaining expressivity. TensorRL (fixed) demonstrates the best capability for representing complex quantum correlations at scale, with its worst initialization often outperforming the best initializations of alternative methods. The consistent 50% maximum success probability for 10-qubit systems provides unprecedentedly efficient circuit discovery for moderately sized molecules, representing a significant advancement for Hamiltonian ground state estimation.

## H.4 TensorRL on CPU

Figure 6: **Comparison of training time and circuit efficiency for TensorRL (fixed CPU/GPU) versus CRLQAS (GPU) and vanilla RL (GPU)**: TensorRL (fixed, CPU) achieves up to 10× fewer gates to chemical accuracy and matches or surpasses GPU-based baselines in training time, enabling efficient and accessible quantum architecture search on standard CPUs. Here CA denotes the chemical accuracy.

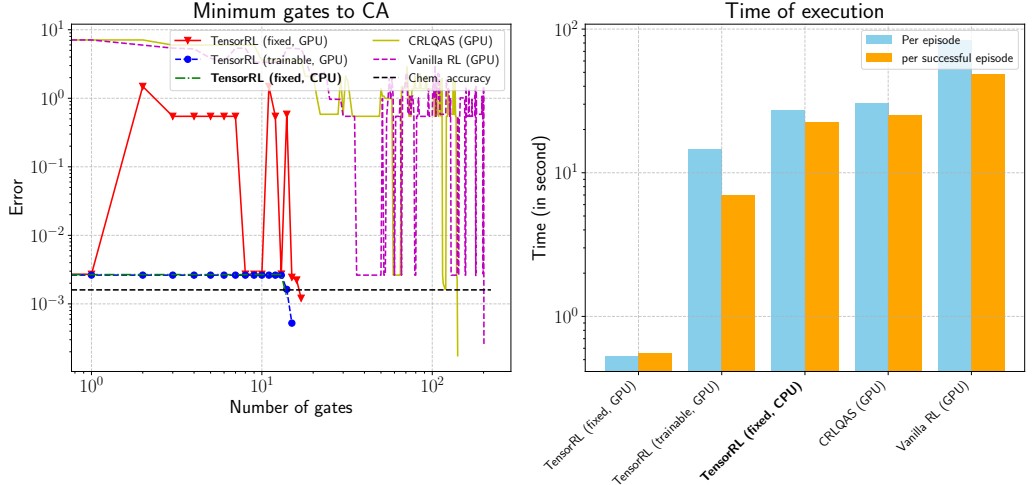

Our experimental results, summarized in Figure 6, demonstrate that TensorRL in trainable mode on CPU achieves training efficiency that is competitive with, and often superior to, established baselines such as CRLQAS and vanilla RL operating on GPU. For quantum architecture search tasks up to 8-qubit, TensorRL (trainable, CPU) consistently matches or outperforms the GPU-based baselines in terms of wall-clock time per episode and per successful episode. Notably, this improvement in computational efficiency does not come at the expense of solution quality: the minimum number of gates required to reach chemical accuracy is comparable to, or slightly better than, those achieved by the baselines. These results establish that RL-QAS, when implemented with TensorRL, is practically trainable on standard CPUs for problem sizes up to 8-qubit, significantly broadening accessibility to high-quality quantum circuit design without reliance on specialized hardware.

## H.5 TensorRL-QAS beyond chemical Hamiltonians

Table 6 presents a comprehensive comparison of TensorRL and recent QAS methods on the 5-qubit Heisenberg model

$$H_{\text{Heis}} = \sum_{i=1}^{n} X_i X_{i+1} + Y_i Y_{i+1} + Z_i Z_{i+1} + \sum_{i=1}^{n} Z_i, \tag{16}$$

and the 6-qubit transverse field Ising model (TFIM at $h = 5 \times 10^{-2}$)

$$H_{\text{TFIM}} = \sum_{i=1}^{n} Z_i Z_{i+1} + h \sum_i X_i, \tag{17}$$

These results demonstrate TensorRL's strong performance in general quantum many-body settings.

Table 6: **Comparison of TensorRL-QAS with baseline QAS methods for the 5-qubit Heisenberg and 6-qubit TFIM models**. Bold: lowest gate count for competitive error. NA: not available.

| Problem | Method | Error | Depth | CNOT | GATE |
|---|---|---|---|---|---|
| 5-Heisenberg | **TensorRL (fixed)** | $7.6 \times 10^{-2}$ | **15** | **5** | **33** |
| | TensorRL (trainable) | $3.7 \times 10^{-4}$ | 46 | 28 | 127 |
| | StructureRL | $2.6 \times 10^{-3}$ | 48 | 33 | 127 |
| | TF-QAS [42] | $1.2 \times 10^{-3}$ | NA | NA | 35 |
| | DQAS (best) [18] | $1.1 \times 10^{-1}$ | NA | NA | 35 |
| | GQAS (best) [86] | $7.1 \times 10^{-4}$ | NA | NA | 35 |
| | GQAS-SSL (best) [86] | $\mathbf{4.0 \times 10^{-6}}$ | NA | NA | 35 |
| 6-TFIM | **TensorRL (fixed)** | $2.4 \times 10^{-4}$ | **13** | **13** | **22** |
| | TensorRL (trainable) | $1.8 \times 10^{-6}$ | 29 | 16 | 111 |
| | StructureRL | $1.8 \times 10^{-6}$ | 34 | 20 | 116 |
| | TF-QAS | $\mathbf{1.4 \times 10^{-12}}$ | NA | NA | 36 |

TensorRL achieves competitive errors with much fewer gates than other approaches, especially for the fixed TN-init variant. For the 6-qubit TFIM, TensorRL finds solutions in circuits as small as 22 gates, an order of magnitude smaller than other competitive methods. The trade-off between solution accuracy and circuit complexity highlights TensorRL's capacity to generate highly compact and efficient circuits for quantum many-body systems outside chemical applications.

## H.6 Scaling beyond 12-qubit

We examine the scalability of TensorRL-QAS and discuss its current limitations. To assess, we consider transverse field Ising models of 15-, 17-, and 20-qubit, with the Hamiltonian defined in Eq. 17 and a magnetic field strength of $h = 10^{-3}$. As established in Section 3 and Section 6, the *TensorRL (fixed)* variant yields the lowest computational overhead per reinforcement learning (RL) episode. Therefore, we adopt this method to evaluate scalability up to 20-qubit across increasing system sizes and different Hamiltonians.

Our results, summarized in Table 7, demonstrate that TensorRL-QAS yields accurate approximations of the ground state energy for the TFIM up to 17 qubits. Specifically, the method improves the energy by up to 21% relative to the initial MPS state, achieving approximation errors of $4.4 \times 10^{-4}$ for the 15-qubit system and $4.8 \times 10^{-4}$ for the 17-qubit. For the standard 20-qubit case with a brickwork operator pool, while the absolute error remains low ($6.7 \times 10^{-4}$), the relative improvement is minimal (1%), indicating a bottleneck due to limited action space and MPS bond dimension. Importantly, when the operator pool is expanded to include two-qubit operators (XX, YY, ZZ) to the already existing single-qubit rotations (RX, RZ, RY) and CX, the improvement on the same 20-qubit TFIM increases to 9%. This clearly demonstrates that a more expressive set of actions enables TensorRL-QAS to scale more effectively and achieve superior ground state approximations for challenging large-scale problems. We believe this performance can be further improved by utilising the operator pool described in ref. [16] or ref. [87].

Table 7: **Analysis of TensorRL-QAS for approximating ground-state energies of 15-, 17-, and 20-qubit transverse-field Ising models (TFIM) with weak magnetic field** ($h = 10^{-3}$). Results demonstrate up to 21% energy improvement over initial MPS states for systems up to 17-qubit (errors $\sim 10^{-4}$), while the 20-qubit case reveals limitations due to restricted bond dimension ($\chi = 2$) and brickwork action space. With an extended operator pool for the 20-qubit case, substantial improvement is recovered.

| Problem | Method | Error | Improvement |
|---|---|---|---|
| 15-TFIM | TensorRL (fixed) | $4.4 \times 10^{-4}$ | 21% |
| 17-TFIM | TensorRL (fixed) | $4.8 \times 10^{-4}$ | 19% |
| 20-TFIM (bottleneck) | TensorRL (fixed TN-init) | $6.7 \times 10^{-4}$ | 1% |
| **20-TFIM (actions: XX, YY, ZZ, RX, RZ, RY, CX)** | TensorRL (fixed) | $6.1 \times 10^{-4}$ | **9%** |

# I   Effect of different MPS to circuit mapping on TensorRL-QAS

To investigate how the properties of the tensor network (TN) initialization influence TensorRL-QAS, we evaluated a range of TN-to-circuit mappings by varying the number of TN layers (1 to 3) as the warm-start for $6 - \text{BEH}_2$, $8 - \text{H}_2\text{O}$, and $10 - \text{H}_2\text{O}$ molecular instances. For each configuration, we recorded the RL-agent's success rate, gate counts, circuit depth, and ground-state error.

Table 8: **Performance of TensorRL-QAS for different TN initialization depths and molecules**. For each molecule, the lowest ground-state error for each group is in bold.

| Molecule | TN layers | Success rate | CNOT | ROT | Depth | Error |
|---|---|---|---|---|---|---|
| $6 - \text{BEH}_2$ | 1 | 0.85 | 5 | 12 | 7 | $\mathbf{6.0 \times 10^{-5}}$ |
| | 2 | 0.84 | 3 | 16 | 8 | $1.2 \times 10^{-4}$ |
| | 3 | 0.85 | **2** | **4** | **4** | $2.4 \times 10^{-4}$ |
| $8 - \text{H}_2\text{O}$ | 1 | 0.14 | 9 | 15 | **6** | $\mathbf{8.9 \times 10^{-4}}$ |
| | 2 | 0.14 | **8** | 11 | 9 | $1.3 \times 10^{-3}$ |
| | 3 | 0.00 | 6 | 14 | 7 | $1.9 \times 10^{-3}$ |
| $10 - \text{H}_2\text{O}$ | 1 | 0.13 | 15 | 17 | **7** | $\mathbf{4.1 \times 10^{-4}}$ |
| | 2 | 0.11 | **13** | **8** | 9 | $5.8 \times 10^{-4}$ |
| | 3 | 0.00 | 18 | 12 | 16 | $1.8 \times 10^{-3}$ |

For $6 - \text{BEH}_2$, both 2- and 3-layer initializations yield high success rates and competitive circuit costs. However, for $8 - \text{H}_2\text{O}$ and $10 - \text{H}_2\text{O}$, increasing TN initialization depth to 3 layers consistently leads to RL failures, while the 2-layer variant maintains reasonable performance. These results indicate that deeper or more complex TN initializations can actually hinder overall algorithmic performance, even within the same TN family, likely due to reduced learnability rather than overfitting. Our findings emphasize that an effective TN initialization strikes a practical balance between expressivity and learnability. Systematic exploration of fundamentally different TN structures and alternative mappings is a valuable avenue for future work.

# J   Performance of TN-compiled PQCs without RL refinement

To isolate the contribution of the TN-init alone, we report ground-state energies for several molecular systems using only pure TN-compiled circuits, without any RL-based refinement. The table below shows the energy error relative to the true reference as a baseline.

This baseline highlights the importance of subsequent RL refinement for achieving chemical accuracy, as the pure TN-compiled circuits alone yield higher errors, especially for larger systems.

Table 9: **Ground state energies and errors for pure TN-compiled PQCs (without any RL refinement)**. The error is relative to the reference ground-state energy.

| Molecule | TN energy | True ground | Error |
|---|---|---|---|
| $6 - \text{BEH}_2$ | $-14.85$ | $-14.86$ | $5.9 \times 10^{-3}$ |
| $8 - \text{H}_2\text{O}$ | $-73.29$ | $-73.29$ | $2.7 \times 10^{-3}$ |
| $10 - \text{H}_2\text{O}$ | $-74.56$ | $-74.56$ | $4.8 \times 10^{-3}$ |
| $12 - \text{LiH}$ | $-7.68$ | $-7.87$ | $1.9 \times 10^{-1}$ |

## K    Scaling analysis: DMRG bond dimension vs RL-agent refinement

In this section we discuss how much the agent refines the performance of the tensor initialization with increasing bond dimension. Tab. 10 provides a systematic comparison of RL optimization time (RL opt. time), time per episode and final accuracy (error) as a function of the initial DMRG bond dimension for a $12 - \text{LiH}$ ground state preparation. The results reveal that increasing the bond dimension in DMRG significantly reduces both the overall TensorRL training time and time per episode, as the higher-quality initialization leads to faster convergence and improved circuit efficiency.

Table 10: **The RL optimization time and accuracy for 12-qubit LiH as a function of initial DMRG bond dimension**. Increasing DMRG bond dimension leads to reduced RL training time and convergence to chemical accuracy.

| DMRG bond dim. $(\chi)$ | RL opt. time (h) | Episode time (s) | Error |
|---|---|---|---|
| 2 | 15.8 | 195 | $7.2 \times 10^{-3}$ |
| 4 | 7.4 | 92 | $1.5 \times 10^{-3}$ |
| 8 | 3.9 | 49 | $1.4 \times 10^{-3}$ |
| 16 | 2.3 | 29 | $1.2 \times 10^{-3}$ |
| 32 | 1.1 | 14 | $1.2 \times 10^{-3}$ |

However, once the DMRG initialization reaches chemical accuracy, further improvements from RL become marginal, indicating that classical methods remain most efficient in this tractable regime. The practical advantage of the RL framework emerges as DMRG scalability breaks down, where quantum circuit construction and RL-based refinement become vital. This scaling analysis clarifies the operational regime of the TensorRL-QAS: as classical tensor methods approach their tractability limits, RL-guided quantum circuit discovery becomes increasingly relevant.

## L    Scalability bottlenecks: Analysis of quantum simulator

The Fig. 7 presents a critical comparison between baseline reinforcement learning-assisted quantum architecture search (RL-QAS) and tensor network (TN) for RL-QAS (TensorRL-QAS) approaches, revealing fundamental scalability challenges in quantum computing. As depicted in the left panel, the baseline RL-QAS method exhibits a severe computational bottleneck where statevector calculation time increases exponentially with both circuit depth and qubit count. For a system with $N = 12$ qubits, computation time reaches approximately 10 seconds at circuit depths approaching 400, while even modest depths with $N = 8$ qubits require significantly less time (approximately 0.1 seconds at depth 200). This exponential scaling represents a fundamental limitation to practical quantum applications.

A crucial consideration for real-world implementation is that reinforcement learning agents typically require multiple episodes to converge to optimal policies. If training necessitates an average of 10,000 episodes, as is common in complex RL tasks, the total computation time for RL-QAS becomes entirely intractable. With each episode requiring up to 10 seconds of statevector calculation for $N = 12$ qubits, the complete training process could require approximately 100,000 seconds (over 27 hours) of computation time solely for statevector calculations. This computational burden effectively precludes scaling to larger, more practical quantum systems where 12+ qubits would be required.

Figure 7: **Time taken by** `Qulacs` **[88] simulator to get the statevector with baseline RL-QAS and TensorRL-QAS methods with varying numbers of qubits and depths of circuit**. The substantial difference in computation time becomes even more critical when considering that training requires approximately 10,000 episodes, making RL-QAS approaches intractable for larger qubit systems.

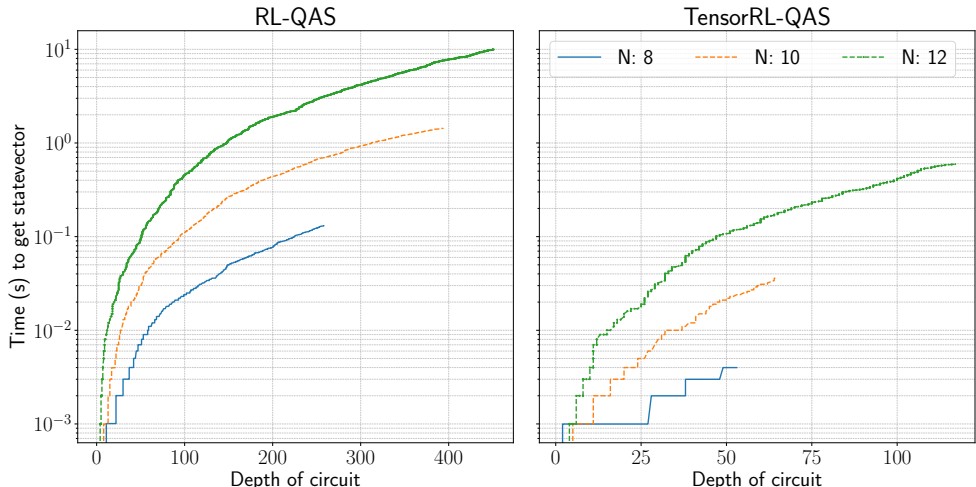

In stark contrast, the right panel demonstrates TensorRL-QAS's superior performance characteristics. TensorRL-QAS requires substantially shallower circuits to achieve comparable results, with maximum depths of only about 100 compared to 400 for baseline methods. More importantly, computation times remain under 0.5 seconds even for $N = 12$ qubits, reducing the projected training time for 10,000 episodes to approximately 5,000 seconds (less than 1.5 hours). This dramatic improvement stems from TensorRL-QAS's innovative "warm start" approach, which leverages matrix product states (MPS) for parameterized quantum circuit (PQC) initialization.

The warm start strategy enables TensorRL-QAS to converge to solutions in significantly fewer steps, creating a computational advantage that increases with qubit count. By effectively mapping the tensor network representation to quantum circuit parameters, TensorRL-QAS takes a step forward to addresses one of the primary obstacles in quantum architecture search i.e. scalability. This initialization method bypasses the exponential growth in statevector computation time that plagues conventional approaches, making TensorRL-QAS substantially more suitable for practical quantum applications requiring larger qubit systems.

The logarithmic scale in the figure clearly illustrates the magnitude of this advantage, with the baseline method showing steep growth curves across all qubit counts. At the same time, TensorRL-QAS maintains relatively modest computation times even as system size increases. These results strongly suggest that tensor network initialization strategies represent a promising direction for overcoming limitations in quantum machine learning approaches, potentially enabling quantum advantage in practical applications that would otherwise remain computationally intractable using conventional methods.

## M   Experimental setup

### M.1   Reward function

We employ the reward function $R$ from [21, 35] with modifications for quantum tasks:

$$R = \begin{cases} 5, & \text{if } C_t < \xi \\ -5, & \text{if } t \geq T_s^e \ \& \ C_t \geq \xi \\ \max\left(\dfrac{C_{t-1} - C_t}{C_{t-1} - C_{\min}}, \ -1\right), & \text{otherwise} \end{cases} \tag{18}$$

where $C_t = \langle 0|U^\dagger(\boldsymbol{\theta})HU(\boldsymbol{\theta})|0\rangle_t$ represents the variational energy at step $t$, $\xi$ is a precision threshold, and $T_s^e$ denotes the episode's step limit. The $\pm 5$ rewards terminate episodes upon success (energy $< \xi$) or failure (timeout with energy $\geq \xi$).

For quantum chemistry tasks, the threshold $\xi$ is typically set to $1.6 \times 10^{-3}$ as it defines the precision such that realistic chemical predictions can be made. The goal of the agent is to obtain an estimated value of $C_{\min}$ within such precision. Whereas, while solving the Heisenberg model, we consider $\xi$ to $10^{-3}$ and for the transverse field Ising model we set it as $10^{-2}$, which are obtained by running SA-QAS [39] on these Hamiltonians.

### M.2 Number of steps per episode

Here we report the number of steps (i.e., maximum number of gates can be added by the RL-agent) per episode for TensorRL-QAS and baseline QAS algorithms to reach convergence to the ground state energy, as summarized in Table 11. The results highlight that TensorRL-QAS consistently

Table 11: **Steps per episode** required for convergence to ground energy.

| Qubits | TensorRL-QAS | CRLQAS | RAQAS |
|--------|--------------|--------|-------|
| 6      | **20**       | 70     | 97    |
| 8      | **20**       | 250    | 300   |
| 10     | **20**       | 350    | 477   |
| 12     | **40**       | 450    | 450   |
| 15     | **60**       | NA     | NA    |
| 17     | **60**       | NA     | NA    |
| 20     | **80**       | NA     | NA    |

achieves convergence with dramatically fewer steps, often by a factor of $3 - 5\times$ or more compared to CRLQAS and RAQAS, across all tested system sizes. This reduction in episode length reflects the benefit of starting from a physically motivated tensor network initialization, substantially accelerating learning and improving computational efficiency in quantum circuit discovery.

Table 12: **The geometry and basis of molecules used in this research**. The coordinates are in Angstrom units.

| Molecule | Geometry | Basis |
|----------|----------|-------|
| 6-BEH$_2$ | H (0,0,-1.33); Be (0,0,0); H (0,0,1.33) | STO-3G |
| 8-H$_2$O | H (-0.02,-0,0); O (0.84,0.45,0); H (1.48,-0.27,0) | STO-3G |
| 8-CH$_2$O | C (0,0,0); H (1.08,0,0); H (-0.23,1.06,0) | STO-3G |
| 10-H$_2$O | H (-0.02,-0,0); O (0.84,0.45,0); H (1.48,-0.27,0) | 6-31G |
| 10-CH$_2$O | C (0,0,0); H (1.08,0,0); H (-0.23,1.06,0) | STO-3G |
| 12-LiH | Li (0,0,0); H (0,0,3.40) | STO-3G |

### M.3 Molecular configuration

Using the Born-Oppenheimer approximation, we fix a finite basis set, in our case STO-3G, for all the molecules except 10-H$_2$O, where we use 6-31G to extend the number of orbitals and to discretize the system. We choose the molecules based on the top 20 molecules provided by `Pennylane` [61], and throughout the paper, we use `Pennylane` along with `OpenFermion` to generate the chemical Hamiltonians in Tab. 12.

### M.4 MPS to PQC conversion gate resources

In Tab. 13 we present the resources in the one-depth brickwork structure required when converting from MPS to parameterized quantum circuit. The resources are quantified by the number of CNOTs,

rotations (ROT) and depth (Depth) of the circuit and in the case of 10-qubit, the bond dimension $\chi$. All the circuits are simulated using `Qulacs` [88].

Table 13: **Gate counts and depth for MPS-to-circuit conversion across different system sizes.** The number of gates increases with bond length $\chi$.

| Qubits | CNOT | Depth | ROT |
|---|---|---|---|
| 5 ($\chi = 2$) | 12 | 27 | 75 |
| 6 ($\chi = 2$) | 15 | 27 | 93 |
| 8 ($\chi = 2$) | 21 | 27 | 129 |
| 10 ($\chi = 2$) | 27 | 27 | 165 |
| 10 ($\chi = 3$) | 26 | 27 | 160 |
| 12 ($\chi = 2$) | 33 | 27 | 201 |
| 15 ($\chi = 2$) | 42 | 27 | 255 |
| 17 ($\chi = 2$) | 48 | 27 | 291 |
| 20 ($\chi = 2$) | 57 | 27 | 345 |

# N  Computed resources

We utilize a system with an `AMD Rome 7H12 CPU` based on `AMD Zen 2 architecture` and an `Nvidia Ampere A100 GPU`. The time of training and approaching the first and minimum error is presented in Tab. 14, all times are noted in hours. For all the configurations, we utilized 2 CPUs, each with a maximum 4GB memory and maximum 40 GB GPU memory.

Table 14: **The training time for different TensorRL-QAS methods across various qubits**. We notice that *TensorRL (fixed)* converges much faster to the desired error than other methods.

| Method / Metric (in hours) | 5q | 6q | 8q | 10q | 12q |
|---|---|---|---|---|---|
| **TensorRL (fixed)** | | | | | |
| Time to min error | 5.88 | 0.05 | 0.91 | 11.22 | 0.58 |
| Time to first success | 5.24 | 0.05 | 0.15 | 0.53 | 0.58 |
| Total train time | 6.42 | 0.13 | 1.48 | 34.82 | 48.00 |
| **TensorRL (trainable)** | | | | | |
| Time to min error | 6.14 | 0.24 | 27.60 | 0.56 | 48.15 |
| Time to first success | 6.14 | 0.02 | 0.40 | 0.56 | 17.07 |
| Total train time | 29.84 | 2.00 | 40.84 | 47.50 | 48.00 |
| **StructureRL** | | | | | |
| Time to min error | 5.42 | 0.66 | 35.47 | 23.92 | 14.86 |
| Time to first success | 5.42 | 0.27 | 1.03 | 23.92 | 4.81 |
| Total train time | 29.84 | 1.91 | 45.75 | 47.69 | 48.00 |

# O  Borderline impact

1. **Accelerating quantum scientific discovery (positive):** TensorRL-QAS enables scalable and efficient quantum circuit design, which can accelerate research in quantum chemistry, materials science, and related domains. This may facilitate the discovery of new molecules, drugs, or materials, with broad societal benefits in medicine, energy, and technology.

2. **Democratizing access to quantum algorithm development (positive):** By significantly reducing computational overhead and enabling CPU-only training for sizable quantum systems, our method lowers barriers to entry for researchers and institutions with limited resources, promoting broader participation in quantum computing research.

3. **Enabling practical quantum advantage (positive):** The approach addresses key bottlenecks hindering the realization of quantum advantage on near-term hardware. Making

variational quantum algorithms more feasible and robust under noise brings practical quantum applications closer to reality, with potential for positive economic and technological impact.

4. **Potential for misuse (negative):** As with any advancement in quantum computing, more efficient quantum circuit discovery could be leveraged for malicious purposes, such as breaking cryptographic protocols or enabling unauthorized access to secure systems. While this work focuses on quantum chemistry and physics applications, the generality of the approach means it could be adapted for less benign purposes.

5. **Environmental considerations (negative):** Although the method reduces computational requirements relative to previous approaches, quantum computing research-especially at scale-remains energy-intensive. Widespread adoption could contribute to increased energy consumption if not managed responsibly, particularly as quantum hardware becomes more prevalent.

**Safeguards and limitations:** The code is released openly with documentation at https://github.com/Aqasch/TensorRL-QAS and appropriate licensing. The work does not introduce high-risk assets such as generative models for disinformation or surveillance. Limitations and future work are discussed, including the need for real-device validation and further exploration of action space expressivity; immediate risks are assessed to be low in the context of the presented research. This work discusses both the potential positive and negative societal impacts, in line with NeurIPS requirements for a balanced and responsible broader impact statement.

