# OpenReview forum: "TensorRL-QAS: Reinforcement learning with tensor networks for improved quantum architecture search"
_NeurIPS.cc/2025/Conference — NeurIPS 2025 poster_

### Official Review · Reviewer_e9Vh · 2025-06-21

**Clarity:** 3
**Significance:** 2
**Originality:** 2
**Rating:** 4
**Confidence:** 5

**Summary:**

The paper proposes using a tensor network simulation of a parameterized quantum circuit to warm start a variational simulation of a quantum chemistry problem.  The variational simulations are performed classically using the qulacs simulator, allowing the systems to be simulated in both closed and noisy environment, on a quantum architecture of their choosing.  The authors claim that this method reduces the circuit depth required to find high quality solutions by a substantial amount (see table 1).

**Questions:**

Why is bond dimension 2 sufficient for these simulations?  I would have expected these simulations to need a bond dimension of $2^5$ - $2^7$  for the 8 qubit case, and more for a larger simulation.

If bond dimension 2 is sufficient for these simulations, why haven't you scaled the simulation to significantly larger system sizes (50+ qubits) when in principle an 8 qubit matrix product state implementation should take a very small amount of memory to store classically?

What tensor network implementation are you using?

How can you guarantee that these results will continue to scale after the point where classical algorithms break down?

**Ethical Concerns:**

["NO or VERY MINOR ethics concerns only"]

**Final Justification:**

The work strikes me as sound, and the authors use a somewhat interesting approach.  It is not entirely clear how substantive the advantage over fully classical workflows will be in full scale simulations, but it seems to have a decent chance of giving some improvement, most likely in accuracy, when a fault tolerant quantum simulator of the appropriate scale is available.

**Limitations:**

There are no apparent negative societal implications of their work.

**Paper Formatting Concerns:**

As mentioned above, a few typos which need to be resolved.

**Quality:**

2

**Strengths And Weaknesses:**

The paper proposes a rather interesting idea, with regards to identifying useful architectures for a variational quantum circuit, but unfortunately they simply do not present enough information for it to be clear that their results are correct, or if they are correct, that they would be generalizable to a system that isn't classically simulable.  I will start with the minor issues, then lead into the bigger issues

There are a decent number of typos in the paper (which is okay, as it is very possible that English is not the first language of the authors).  Regardless of the reason, the paper needs to go through another round of copy editing.  A few examples that come to mind are for example on lines 52,53, where they say "8 qubit" and "4 qubit", when it would make more sense to say "8 qubits" or "4 qubits".  There are other typographic issues throughout the paper, but unfortunately I don't have the time to list each of them for easy correction.  I will note that in equation 2, the sum over the indices should be $i_1, i_2, ..., i_n \in (0, 1)^{n}$ and $k \notin (1,n)$ rather than $k \neq (1,n)$ (note that the reviewer portal won't let me use escaped curly braces, so I use (...) to indicate a set) since this is an issue of mathematical correctness.  They also don't define $D$, which I suspect is the circuit depth on the quantum device, but it could be something else.

Assuming that their methods are correct, it is not entirely clear to me how well these methods would scale if they were only applied to a regime where fully classical workflows would be sufficient, but it could yield a slight improvement, functioning effectively as a memory-time tradeoff for systems just out of reach of classical methods currently.

---

> ### Author Rebuttal · Authors · 2025-07-27
>
> We sincerely thank you for the valuable feedback on the manuscript, and appreciate that you agree on the interesting idea motivating our work. In the following we systematically provide the response to your comments.
>
> ### **There are a decent number of typos in the paper (which is okay, as it is very possible that English is not the first language of the authors). Regardless of the reason, the paper needs to go through another round of copy editing...**
>
> We acknowledge the presence of typos and missing definitions, we will perform a thorough proofreading of the manuscript to correct all exposition issues.
>
> ### **Why is bond dimension 2 sufficient for these simulations? I would have expected these simulations to need a bond dimension of $2^5$-$2^7$ for the 8 qubit case, and more for a larger simulation**
>
> You have raised a valid concern regarding the use of a low bond dimension, and we now clarify that this choice is in fact intentional. For the relatively small quantum chemistry problems considered, a larger bond dimension would be sufficient to produce solutions that are already within chemical accuracy of the ground state energy. In this sense, the low bond dimension is not a limitation but rather an intentional choice: using a larger bond dimension would allow us to solve the problem classically, without the need of any additional RL-based quantum computing resource.
>
> We emphasize that the current study is a proof-of-concept intended to validate the proposed combination of TN and RL-QAS, and we start testing it on small systems due to the heavy computational requirements needed to train RL agents (see the next discussion). Applying the same methodology on a larger number of qubits would probably require larger bond dimensions in the MPS/DMRG, which is consistent with your expectation.
>
> *We also clarify a technical point:* for an MPS (with open boundary conditions), the maximum required bond dimension to represent an arbitrary n-qubit state exactly is $2^{⌊n/2]}$. For n=8, this means that a bond dimension of $2^4$ is the theoretical maximum needed to express any arbitrary state exactly, not $2^5$ or $2^7$ as suggested. We have clarified this in the revised manuscript to avoid confusion.
>
> ### **If bond dimension 2 is sufficient for these simulations, why haven't you scaled the simulation to significantly larger system sizes (50+ qubits) when in principle an 8 qubit matrix product state implementation should take a very small amount of memory to store classically?**
>
> Thank you for the interesting question. We want to clarify that the main computational bottleneck in our simulations does not lie in the MPS/DMRG simulations (it is inexpensive as long as the bond dimension is not too big), but in the RL training phase. Each training episode requires simulating and optimizing a parameterized quantum circuit, and computing its expectation value multiple times equal to the number of (episodes X steps). Already for ~12 qubits, this becomes computationally intensive due to the number of episodes and steps required for convergence/solving the task. This currently limits our ability to explore system sizes beyond ~20 qubits, as already at these sizes training the RL agent can take approximately 48 hours, which is the maximum runtime available on the University computing cluster which is currently available to us. Future work will address this by further alleviating the computational requirements of the RL training process, and investigating more performant quantum circuit simulators.
>
> Simulating quantum circuits is computationally very hard in general, which is indeed the reason why we propose to combine a classical warm-start with an RL agent that completes the task by accessing quantum resources directly. But since actually accessing quantum hardware is hard because of limited access by providers, we are bound to do simulations of quantum circuits. Finally, to the best of our knowledge, we remark that our work reports the largest RL-based QAS simulation to date with 20 qubits.
>
> ### **What tensor network implementation are you using?**
>
> We use the *Quimb Python library* for all tensor network computations and MPS manipulations, including DMRG simulations and mapping to quantum circuits.
>
> ### **How can you guarantee that these results will continue to scale after the point where classical algorithms break down?**
>
> We acknowledge this is a central open question for all hybrid classical-quantum algorithms, including ours. Like most near-term quantum algorithms, we cannot provide theoretical guarantees that our TensorRL-QAS approach will outperform classical methods at much larger scales. However, we believe our strategy of maximizing classical pre-processing with tensor networks not only will help reduce the quantum resources required, which is especially important for the near- and mid-term regime, but also very much aligns with the current expectation that useful quantum advantage will be achieved with a tight combination of quantum and classical high-performance computing. Overall, we are confident that TensorRL-QAS provides a necessary starting point for future investigations on larger sizes, where a quantum advantage may manifest. We have added explicit discussion of these considerations and limitations in the revised manuscript.
>
> We hope these addresses your concerns and questions and further helps you to re-evaluate our paper. Kindly let us know if you have any further comments or/and questions.

---

> > ### Comment · Reviewer_e9Vh · 2025-08-05
> > **Thank you for your response - further clarification requested**
> >
> > With regards to your clarification: you are correct it would scale proportionally to $2^{\lfloor{n/2}\rfloor}$.  In my mind, I had flipped where the division by 2 belonged.  That said, the broader point still stands from my original question that the bond dimension would scale proportional to the system size for an approximate simulation.  Now I understand the rationale of the low bond dimension, thank you.
> >
> > However, if a larger bond dimension would trivially solve the problem within chemical accuracy, then it seems to me that the reinforcement learning approach is immediately inferior to simply increasing the bond dimension for the system sizes on which it is considered, unless a scaling study shows otherwise.  This is based on your procedure in lines 122 - 128, where implicitly, if DMRG has converged, you terminate after step 1.  As you mention, the point is to see how it will behave in the regime where step 1. fails, but it is still unclear to me from this work whether steps 2 and 3 are feasible in the regime where step 1 does not converge if the optimization is already hitting HPC bottlenecks on systems of this size.  Of course it would be clearly unreasonable of me to ask you to show that it works in the regime where a fault tolerant quantum computer would be required, but can you show how the optimization time scales as the bond dimension of the initial DMRG increases (for example just from bond dimension between 2^1 - 2^6 for the 12 qubit instance)?  If the scaling of the runtime of the RL optimization doesn't improve markedly as the bond dimension increases, then it would indicate that this approach wouldn't be able to reach very far beyond the regime that DMRG is already able to access, making it a very niche use case at best.  The argument could be made that this study is a proof of concept that would become accessible when a fault tolerant quantum computer exists, but if that is the case, a reasonable estimate of runtimes for computing the expectation values in each epoch on the quantum computer would be necessary (likely a range of simulation times due to the variety of different proposed architectures).
> >
> > In summary, I am still left unclear in what regime (NISQ, Small Scale Fault Tolerant, Intermediate Scale Fault Tolerant) these results would be an improvement beyond the current limits of DMRG due to the time to train the model and the sampling overheads that would be required as we surpass the regime where these kinds of simulations could be carried out classically.  I would appreciate further clarification. Thank you again for your response.

---

> ### Author Response · Authors · 2025-08-05
>
> Thank you very much for your thoughtful comments and for pushing us to examine the scaling of our RL optimization with respect to the initial DMRG bond dimension. We have now performed additional benchmarks specifically as you suggested, on a 12-qubit LiH molecule, using 100 RL episodes per run (this is a result with just one seed, in the camera ready version we will have the same outcome with multiple initilizations of the neural net). For each bond dimension, our target was to reach an energy below that from simulated annealing (2.5×10⁻² Hartree). Our updated results are included in the following table:
>
> | Bond dimension | Total optimization time to minima (minutes) | Total RL steps to minima | Time per episode (seconds) |
> |:--------------:|:-----------------------:|:---------------:|:--------------------:|
> | $2$              | 298.68                  | 4050            | 212.77               |
> | $2^2$          | 208.06                  | 3550            |  193.95    |
> | $2^3$              | 185.79                  | 3000            | 178.06               |
> | $2^4$             | 141.35                  | 2750            | 109.00               |
> | $2^5$             | 97.61                   | 2300            | 105.49               |
> | $2^6$             | 24.03                   | 758             | 29.56                |
>
> Our results show that both the optimization time and the episode time decline noticeably as the initial DMRG bond dimension increases. We think this is a natural consequence of providing a better starting point for RL optimization. That said, we completely agree with your main point: for these modest system sizes, where DMRG with higher bond dimension is already feasible, increasing DMRG bond dimension is both straightforward and effective. We do not see a compelling speed or efficiency advantage for RL under these circumstances. In our view, RL’s potential value might be in scenarios with stronger resource constraints either larger or more entangled systems.
>
> Additionally, we would like to clarify how our timing data relates to the question of estimating quantum runtime. The "estimate of runtimes for computing the expectation values" reported in the table above reflects the total simulation time required to complete one RL episode on our classical simulator. Each episode in our setup is divided into 50 epochs, where in each epoch, expectation values must be computed to update the RL policy. Thus, as an estimate, the time required to compute the expectation values in a single epoch on our simulator corresponds to "time per episode" divided by 50. For example, in the $2^4$ bond dimension case, this amounts to roughly 2.18 seconds per epoch on our simulator.
>
> While this episode timing is specific to our classical simulation environment and does not directly translate to quantum hardware, it provides an indirect indication of the runtime required per epoch for expectation value estimation, which we hope is helpful for benchmarking and planning for future quantum implementations.
>
> **Regarding the question of computational bottlenecks** we would like to clarify that we are using a relatively modest HPC setup: each run was carried out with a single A100 GPU, 2 CPU cores, and 4GB RAM per core. For the system sizes presented here (up to 12 qubits), most runs complete well within the requested wall time. However, achieving higher accuracy-such as reaching error below chemical accuracy (1.6×10⁻³)-can require considerably more resources, and in some cases the wall time may approach the 48-hour maximum allowed on our available cluster. Thus, the computational cost in practice depends not just on system size, but also on the desired accuracy threshold. Moving forward, we plan to further reduce these computational requirements and explore more efficient quantum circuit simulators.
>
> **With respect to your summary question on applicable regimes:** Based on our results, we do not find an advantage for RL over DMRG in the currently accessible, classically tractable regime. The RL approach is mainly motivated by scenarios where DMRG becomes infeasible due to system size or entanglement-potentially relevant for future quantum hardware (NISQ or fault-tolerant), though we acknowledge that the sampling overhead and training costs on such devices are still an open research challenge. At this stage, our work serves as a proof of concept and benchmark; further investigation is needed to pinpoint the hardware requirements and practical regimes where RL could provide clear benefits beyond classical methods. We appreciate your emphasis on this important point.
>
> We hope that our response clarifies your main concerns. Kindly let us know if you have any more comments or suggestions.

---

> > ### Comment · Reviewer_e9Vh · 2025-08-05
> > **Thank you for the response: Probably final question before I submit final review**
> >
> > Thank you for the well thought out response. I am very impressed you managed to get that scaling study run so quickly.  Hopefully it wasn't too late of a night, but I think it adds a lot.
> >
> > This fits rather in-line with what I expected, mainly that as the bond dimension grows closer to the full size, that the optimization time decreases, implying that this method constitutes a memory-time tradeoff.  To describe my understanding of it, an initial solution is found in a lower precision subspace (DMRG with bond dimension lower than $2^{\lfloor{n/2}\rfloor}$), then the reinforcement learning step is using the quantum simulator to perform a local optimization in the full precision space where optimization may prove challenging without a good starting point.
> >
> > If that understanding is correct, then it seems that when using this method with an arbitrary quantum simulator, the optimization space is going to scale with some polynomial of 1. the number of qubits, 2. the ansatz depth, 3. The ratio $2^{\lfloor{n/2}\rfloor}/\chi$, and 4. the required precision (due to both the optimizer and the precision requirements of the quantum simulator, whether classical or shot complexity from a quantum device).  This implies to me that the optimization complexity as the system size scales will prove to be a bottleneck, likely in the same regime where bond dimension on a large scale DMRG simulation will cause a bottleneck for a purely classical approach.  This says to me that it will likely only yield a slight improvement in the regimes where you would need to use this algorithm because you could no longer represent a sufficiently high bond dimension.
> >
> > If you agree with all of this, I will update my review as follows:
> >
> > Quality: 1 -> 2 (upon re-reading I think I was a little bit too harsh with my initial score)
> >
> > Clarity: 2 -> 3 (based on the changes in language to emphasize the reasoning for the low bond dimension used in the DMRG simulation)
> >
> > Significance 2 -> 2 (keeping the same, because I am still not fully convinced that this will allow access to many problems of interest that cannot already be handled by DMRG, independent of the underlying simulator)
> >
> > Originality 2 -> 2
> >
> > Final score: 2 -> 4. Borderline Accept
> >
> > In case you disagree with any of what I have said here, I will leave my final review open for now.  You have done a great job of responding to my questions/issues so far, but I expect this is likely to remain my final score unless I am missing a large detail that negates my concerns about scalability.  Thank you!

---

> ### Author Response · Authors · 2025-08-06
> **Thank you for your kind response and contributions**
>
> Thank you very much for your careful and thoughtful review. We are grateful for your statement regarding the memory-time trade-off in our approach and your detailed analysis of the key scaling considerations. Your observations regarding how the optimization complexity depends polynomially on system parameters are consistent with our own findings, and have helped us reflect more deeply on the limitations of our method.
>
> We fully agree that these factors, in practice, present significant constraints on scalability. As you rightly point out, the memory-time trade-off enables us to reach useful approximations with lower classical resources initially, but it does shift the primary bottleneck to the subsequent quantum optimization stage. While our results suggest that this hybrid strategy can offer practical benefits in select cases, particularly when classical DMRG requires very large bond dimensions—we recognize that these improvements are often modest and highly dependent on the specific system under study. We agree this does not represent a universal advance over classical DMRG.
>
> Taking your valuable feedback into account, we will be careful to make these trade-offs and scaling limitations more explicit in the camera-ready version. In particular, we plan to adjust the title to use “enhanced” instead of “scalable” to more accurately convey the nature of our contribution, as you have suggested. We feel this change better reflects both the spirit and the scope of the work.
>
> We sincerely appreciate your constructive comments, which have been invaluable in helping us better articulate the contributions of our work and clearly define its limitations. We are very grateful for your updated scores.

---

> ### Public Comment · ~Yijun_Wang6 · 2025-12-01
>
> Responsible reviewer！Kind reviewer！

---

### Official Review · Reviewer_AdE9 · 2025-06-25

**Clarity:** 2
**Significance:** 3
**Originality:** 3
**Rating:** 4
**Confidence:** 3

**Summary:**

The paper introduces TensorRL-QAS to address the scalability challenges in Quantum Architecture Search (QAS). Specifically, the authors first employ the classical Density Matrix Renormalization Group (DMRG) algorithm to obtain a good Matrix Product State (MPS) approximation of the target state. This MPS is then mapped into a Parameterized Quantum Circuit (PQC) to serve as a "warm-start" initialization. Finally, a Reinforcement Learning (RL) agent sequentially adds quantum gates to this initial circuit to further optimize its structure and approach the true ground state.

This "TN warm-start" strategy significantly narrows down the RL search space, leading to a substantial reduction in the number of function evaluations and training time. Experimental results on molecular simulation problems up to 12 qubits demonstrate that this method, compared to baseline approaches that start from scratch, can generate more compact circuits with lower or comparable errors, showing superiority in both noiseless and noisy environments.

**Questions:**

1. Your method benefits from a high-quality initial circuit generated by DMRG. To better isolate the value of RL, could you add a "TN-VQE" baseline? This would involve using the COBYLA optimizer directly to optimize the newly added gates, without any RL framework. This would help distinguish the contributions of "a good starting point" versus "a good search strategy".
2. Mapping the MPS to a single-layer brickwork circuit is a strong assumption. How much does this choice affect the final performance? Have you explored other mapping schemes, such as generating deeper or differently structured initial circuits? Could the current results be overstated due to this highly optimized initial structure?
3. In the fixed TN-init model, the initial circuit parameters are frozen. The RL agent's task is essentially to find an optimal solution for a new, shallower circuit. How does the search difficulty of this new problem compare to the original one?

**Ethical Concerns:**

["NO or VERY MINOR ethics concerns only"]

**Final Justification:**

If the authors commit to thoroughly revising the scalability claims across the entire paper to better reflect their experimental scope, I am willing to raise my score to borderline accept.

**Limitations:**

yes

**Quality:**

2

**Strengths And Weaknesses:**

Strengths:
1. The problem addressed—the scalability bottleneck of RL in QAS—is a core challenge in the current Noisy Intermediate-Scale Quantum (NISQ) era.
2. Combining the physical insights of tensor networks (specifically DMRG/MPS) with reinforcement learning is a novel approach.

Weaknesses:
1. The performance boost of TensorRL-QAS is likely attributable to its high-quality starting point from DMRG, rather than the superiority of the RL agent's search strategy. The paper conflates the benefits of a "better initialization" with those of a "better search algorithm." A fairer baseline would involve: 1) Directly optimizing the DMRG-generated initial circuit with a classical optimizer (like COBYLA) without RL. 2) Having all methods start from the same ansatz to see which RL strategy finds the optimal structure more effectively.
2. A core promise of QAS is to discover novel, counter-intuitive, and efficient quantum circuit structures. However, this work maps the MPS to a single-layered brickwork structure, which is a very strong constraint. While hardware-friendly, it drastically shrinks the search space, potentially defeating the purpose of QAS. The claimed "scalability" might be a byproduct of this extreme space simplification. The mere 1% improvement for the 20-qubit system in Appendix E.6 suggests this MPS structure may struggle to represent entanglement in more complex systems, casting doubt on its generality and true scalability.
3. The best-performing model, TensorRL (fixed TN-init), uses a fixed initial circuit, leaving very limited room for RL to operate. For such a "fine-tuning" task, is a complex RL framework necessary? Why not simply append a few layers of a standard ansatz to the initial circuit and optimize all parameters at once with a classical optimizer? The paper lacks a discussion of this simpler alternative, making the necessity and contribution of RL questionable.
4. The title argues for scalability, yet the experiments are demonstrated up to 20 qubits. In the current stage, "scalable" is more appropriately used for systems approaching 50+ qubits, making the claim seem premature.

---

> ### Author Rebuttal · Authors · 2025-07-27
>
> Thank you very much for your thoughtful feedback. We appreciate your acknowledgment of the novelty in integrating tensor network methods and reinforcement learning. In the following we systematically address your concerns:
>
> ### **...To better isolate the value of RL, could you add a "TN-VQE" baseline? This would involve using the COBYLA optimizer directly to optimize the newly added gates, without any RL framework...**
>
> Thank you for your insightful suggestion. In response, we have added the "TN-VQE" baseline to our results table, as shown below. This baseline utilizes the DMRG-initialized circuit as a starting point and employs the classical COBYLA optimizer to optimize each newly added gate, without RL framework.
>
> | Molecule      | Method                                      | Error                | Depth  | CNOT   | ROT      |
> |---------------|---------------------------------------------|----------------------|--------|--------|----------|
> | 6-BEH₂        | **TensorRL (fixed TN-init)**                | 6.8×10⁻⁵            | **7**  | **5**  | **12**   |
> |               | TensorRL (trainable TN-init)                | 6.0×10⁻⁵            | 33     | 22     | 96       |
> |               | $\vdots$                            | $\vdots$        | $\vdots$     | $\vdots$    | $\vdots$       |
> |               | TN-VQE                                      | 5.4×10⁻³            | 10     | 8      | 12       |
> | 8-H₂O         | **TensorRL (fixed TN-init)**                | 8.9×10⁻⁴            | **6**  | **9**  | **15**   |
> |               | TensorRL (trainable TN-init)                | 2.0×10⁻⁴            | 36     | 30     | 146      |
> |               | $\vdots$                            | $\vdots$        | $\vdots$     | $\vdots$    | $\vdots$       |
> |               | TN-VQE                                      | 2.7×10⁻³            | 10     | 14     | 6        |
> | 10-H₂O        | **TensorRL (fixed TN-init)**                | 4.1×10⁻⁴            | **17** | **15** | **17**   |
> |               | TensorRL (trainable TN-init, χ=3)           | 6.7×10⁻⁴            | 33     | 34     | 168      |
> |               | TensorRL (trainable TN-init)                | 7.1×10⁻⁴            | 35     | 48     | 173      |
> |               | $\vdots$                            | $\vdots$        | $\vdots$     | $\vdots$    | $\vdots$       |
> |               | TN-VQE                                      | 1.1×10⁻³            | 26     | 51     | 14       |
> | 12-LiH        | **TensorRL (trainable TN-init)**            | **1.0×10⁻²**        | 31     | 37     | 203      |
> |               | TensorRL (fixed TN-init)                    | 2.4×10⁻²            | **15** | **30** | **9**    |
> |               | $\vdots$                            | $\vdots$        | $\vdots$     | $\vdots$    | $\vdots$       |
> |               | TN-VQE                                      | 8.6×10⁻²            | 33     | 81     | 29       |
>
> The table show that TN-VQE consistently underperforms the RL-based approaches, both in terms of solution accuracy (error) and circuit efficiency (depth, CNOT, and ROT). Notably, for the 6-, 8-, and 12-qubit molecules, TN-VQE is unable to find the ground state with comparable accuracy to the RL-based methods, and in some cases fails to solve the problem. For the 10-qubit, TN-VQE does find the ground state, but the achieved accuracy remains significantly worse than other strategies. This demonstrate that having a high-quality initialization alone is insufficient; the RL framework provides a substantial advantage by better guiding the circuit construction and parameter optimization. Thus, our RL approach is not merely leveraging a good starting point, but also substantially improves the search strategy, enabling the discovery of circuits that are both more accurate and more efficient. We will update the **Table  1** with these results in the **Results** section.
>
> ### **...How much does this choice affect the final performance? Have you explored other mapping schemes, such as generating deeper or differently structured initial circuits? Could the current results be overstated due to this highly optimized initial structure?**
>
> Thank you for raising this important point. Due to time constraints and substantial code changes required for generating differently structured initial circuits, we were unable to explore alternative circuit structures in this work. Instead, we focused on varying the number of TN layers (1-3) to increase the circuit depth within the same mapping, and evaluated across 6-, 8-, and 10-qubit cases. We acknowledge this limitation and plan to investigate different TN circuit structures in future. The RL performance and circuit parameters are in the table below:
>
> | Qubit count | TN layer | Success rate | CNOT | ROT | Depth | Error     |
> |-------------|----------|--------------|------|-----|-------|-----------|
> | 6           | 1        |  0.8478       | 5    | 12  | 7     | **0.00006**   |
> |            | 2        |  0.8418       | 3    | 11  | 5     | 0.00012   |
> |            | 3        |  0.8486       | **2**    | **4**   | **4**     | 0.00024   |
> | 8           | 1        |  0.1423       | 9    | 15  | **6**     | **0.00089**   |
> |            | 2        |  0.1398       | **8**    | **11**  | 9     | 0.00133   |
> |            | 3        |  0.0000       | 6    | 14  | 7     | 0.00196   |
> | 10          | 1        |  0.1288       | 15   | 17   | 17     | **0.00041**   |
> |           | 2        |  0.1097       | **13**   | **8**   | **9**     | 0.00058   |
> |           | 3        | 0.0000       | 18   | 12  | 16    | 0.00178   |
>
> For 6 qubits, both 2- and 3-layers TN initializations perform well, yielding similarly high success rates and competitive circuit costs; however, for 8- and 10-qubit, increasing the TN initialization depth to 3-layers actually causes the RL framework to fail (resulting in zero successful episodes), while the 2-layer initialization maintains reasonable performance. This indicates that deeper or more complex initial circuits, in this brickwork pattern, can hinder rather than overstate algorithmic performance. Thus, our results suggest that the method’s success is not simply due to an overly optimized or "overfitted" initial structure; rather, our chosen TN initialization offers a practical balance between expressivity and learnability. While further exploration of diverse and deeper circuit mappings is a valuable direction for future work, these findings demonstrate that our current initialization does not artificially inflate our results. We will include these additional results and analyses in the **appendix** in revised manuscript.
>
> ### **...How does the search difficulty of this new problem compare to the original one?**
>
> Thank you for your important question. We would like to clarify that the TN initialization provides a warm start for the RL agent by supplying good initial parameters, potentially allowing more efficient convergence by helping avoid less promising regions of the optimization landscape. However, since the underlying Hamiltonian, whether for quantum chemistry, TFIM, or Heisenberg, remains unchanged, the fundamental computational complexity of the ground state search (which is QMA-hard [[Kempe, J., Kitaev, A., & Regev, O. (2006). The complexity of the local Hamiltonian problem. SIAM Journal on Computing, 35(5), 1070-1097.]) is not altered; thus, while TN-initialization can make the process more practical, it does not change the intrinsic difficulty of the original problem. We hope this addresses the impact and intent of the TN-init method in our work. We will clarify this in the **Methods** section.
>
> ### **..The mere 1% improvement for the 20-qubit system in Appendix E.6 suggests this MPS structure may struggle to represent entanglement in more complex systems, casting doubt on its generality and true scalability.**
>
> Thank you for raising such a crucial question. It should be noted that the improvement (compared to the initial TN energy) depends on action space construction, the bond dimension in DMRG and number of layers in the mapping. Here to address your concern instead of considering (*RX*, *RY*, *RZ*, *CX*) as action space we add more expressive gates such as (*XX*, *YY*, *ZZ*) to the action space which enhances the improvement from 1% to 9%:
> | Problem                 | Method                                  | Error           | Improvement |
> |-------------------------|-----------------------------------------|-----------------|-------------|
> | 20-TFIM (bottleneck)    | TensorRL (fixed)                | $6.7 \times 10^{-4}$      | 1%          |
> | **20-TFIM (action space: *XX*, *YY*, *ZZ*, *RX*, *RZ*, *RY*, *CX*)** | TensorRL (fixed)                | $\mathbf{6.1\times10^{-4}}$                | **9%**      |
>
> Hence the bottleneck of not improving in the ground state preparation task can be addressed by a well defined action space construction. In future work we plan to explore other factors mentioned above to achieve a greater improvement. We will update the **Table** in **Appendix E.6** in the revised version of the manuscript.
>
> ### **The title argues for scalability, yet the experiments are demonstrated up to 20 qubits. In the current stage, "scalable" is more appropriately used for systems approaching 50+ qubits, making the claim seem premature.**
>
> Thank you for your feedback on our manuscript title. We appreciate your point that the original use of "scalable" may be premature. To more accurately reflect the current contribution we have revised the title to:
>
> *"TensorRL-QAS: Reinforcement learning with tensor networks for enhanced quantum architecture search"*
>
> We believe, this new title places emphasis on the methodological advancements our approach offers to quantum architecture search.
>
> We hope these modifications are sufficient and addresses all your concerns. Kindly let us know if you have any more comments, questions or suggestions.

---

> > ### Author Response · Authors · 2025-08-05
> >
> > Dear reviewer,
> >
> > We thank you again for your thoughtful and detailed questions regarding our work, particularly on the inclusion of TN-VQE baseline, the influence of circuit mapping choices, and the complexity of the search task in the fixed TN-init scenario. We found your suggestions highly valuable and have incorporated additional experiments and analyses as thoroughly as possible below which we will include in the camera ready version.
> >
> > We would be truly grateful if you could review our updates and further evaluate our work. Thank you again for your time and consideration.
> >
> > Warm regards,

---

### Official Review · Reviewer_D2Gn · 2025-07-02

**Clarity:** 3
**Significance:** 3
**Originality:** 3
**Rating:** 4
**Confidence:** 4

**Summary:**

The paper combined three proposed ingredients into the novel quantum-architecture search pipeline : (i) DMRG-based tensor-network ground-state preparation, (ii) an MPS-to-quantum-circuit compiler, and (iii) reinforcement-learning optimization using Double-DQN agent to find additional gates that better improve the accuracy. Empirical results on simulated quantum-chemistry Hamiltonians show improved convergence (time-to-accuracy and total gate count) over a baseline RL-QAS without warm starts.

**Questions:**

Can the proposed method be extended to larger systems beyond 12 qubits?


What is the challenge to present practical experimental results using the proposed method?

 Why is RL especially useful for the given task?

**Ethical Concerns:**

["NO or VERY MINOR ethics concerns only"]

**Final Justification:**

I feel the work includes interesting contributions as I mentioned in my review comments. I agree with the other reviewers especially regarding the scalability. My final score is based on these contributions of this paper and its limitation on scalability and applicability,

**Limitations:**

Yes

**Quality:**

3

**Strengths And Weaknesses:**

Strengths

1) This paper combined tensor networks with reinforcement learning to tackle quantum architecture search, which is a highly interesting and promising direction. TensorRL-QAS achieves up to a 10-fold reduction in CNOT count and circuit depth compared to baseline methods, while maintaining or even surpassing chemical accuracy.

2) The manuscript is well-written and the three-stage workflow is easy to follow; implementation details, e.g., network sizes, optimizer hyper-parameters, simulator settings are explicit, which facilitate easy reproduction.

3) On different molecules tested in Table 1, the proposed method consistently accelerates learning and shortens circuits relative to the baseline methods.

4) Simulation results on both noiseless and noisy situations verify the robustness of this method.


Weakness and suggestions for improvement

1) The simulations are limited to 12 qubits, whereas the referenced work [12] reports results on systems with up to 100 qubits. A systematic scaling study on larger qubit numbers would significantly strengthen the paper and demonstrate the broader applicability of the proposed method.


2) The authors have demonstrated some results about noisy cases, while it would be useful to provide some experimental results on real quantum device to demonstrate its practical use.


3) Since the method includes three parts, the author might implement a more detailed ablation study to investigate the contributions of all the three parts. For example, the authors may consider to include the results of a pure TN-compiled circuit without RL refinement.

4）It would be useful to broaden the baseline pool: add recent evolutionary-search and gradient-based QAS methods.

---

> ### Author Rebuttal · Authors · 2025-07-27
>
> Thank you very much for your thoughtful and positive feedback. We are pleased that you find TensorRL-QAS interesting and promising. Moreover we appreciate your positive remarks on the clarity, reproducibility, and robustness. In the following we systematically address all your questions and suggestions.
>
> ### **Can the proposed method be extended to larger systems beyond 12 qubits?**
>
> Thank you for the question. In the current article in appendix we show that TensorRL can be utilized upto 20 qubits, we present these results explicitly in the table below:
>
> | Problem                 | Method                                  | Error           | Improvement |
> |-------------------------|-----------------------------------------|-----------------|-------------|
> | 15-TFIM                 | TensorRL (fixed TN-init)                | 4.4 × 10⁻⁴      | 21%         |
> | 17-TFIM                 | TensorRL (fixed TN-init)                | 4.8 × 10⁻⁴      | 19%         |
> | 20-TFIM (bottleneck)    | TensorRL (fixed TN-init)                | 6.7 × 10⁻⁴      | 1%          |
> | **20-TFIM (extended action space: *XX*, *YY*, *ZZ*, *RX*, *RZ*, *RY*, *CX*)** | TensorRL (fixed TN-init)                | 6.1 × 10⁻⁴                | **9%**      |
>
> The improvement is over the TN initialization energy.
>
> ### **What is the challenge to present practical experimental results using the proposed method?**
>
> The main challenge of our current approach is the computational bottleneck in the RL training phase. More specifically, each episode requires simulating and optimizing a parameterized quantum circuit, and computing its expectation value multiple times (equal to the number of steps i.e. the maximum depth of the circuit set as a hyperparameter). Already for ~12 qubits, this becomes computationally intensive due to the number of episodes required for convergence/solving the task and the increasing number of query to the quantum simulator.
>
> ### **Why is RL especially useful for the given task?**
>
> We thank the reviewer for this valuable question. To show the effectiveness of the RL agent we conduct an experiment where instead of utilizing an RL algorithm to choose and optimize quantum gates we use a hardware-efficient ansatz (HEA) after the fixed TN-init (namely **TN-VQE**). Each gate of the HEA is explicitly optimized by COBYLA optimizer individually, the results are summarized below:
>
> | Molecule      | Method                                      | Error                | Depth  | CNOT   | ROT      |
> |---------------|---------------------------------------------|----------------------|--------|--------|----------|
> | 6-BEH₂        | **TensorRL (fixed TN-init)**                | 6.8×10⁻⁵            | **7**  | **5**  | **12**   |
> |               | TensorRL (trainable TN-init)                | 6.0×10⁻⁵            | 33     | 22     | 96       |
> |               | TN-VQE                                      | 5.4×10⁻³            | 10     | 8      | 12       |
> | 8-H₂O         | **TensorRL (fixed TN-init)**                | 8.9×10⁻⁴            | **6**  | **9**  | **15**   |
> |               | TensorRL (trainable TN-init)                | 2.0×10⁻⁴            | 36     | 30     | 146      |
> |               | TN-VQE                                      | 2.7×10⁻³            | 10     | 14     | 6        |
> | 10-H₂O        | **TensorRL (fixed TN-init)**                | 4.1×10⁻⁴            | **17** | **15** | **17**   |
> |               | TensorRL (trainable TN-init, χ=3)           | 6.7×10⁻⁴            | 33     | 34     | 168      |
> |               | TensorRL (trainable TN-init)                | 7.1×10⁻⁴            | 35     | 48     | 173      |
> |               | TN-VQE                                      | 1.1×10⁻³            | 26     | 51     | 14       |
> | 12-LiH        | **TensorRL (trainable TN-init)**            | **1.0×10⁻²**        | 31     | 37     | 203      |
> |               | TensorRL (fixed TN-init)                    | 2.4×10⁻²            | **15** | **30** | **9**    |
> |               | TN-VQE                                      | 8.6×10⁻²            | 33     | 81     | 29       |
>
> This reveals that without an RL training we can optimize the DMRG-initialized circuit to a certain extent but it consistently underperforms the RL-based approaches, both in terms of solution accuracy (error) and circuit efficiency (depth, CNOT, and ROT counts). Notably, for the 6-, 8-, and 12-qubit molecules, TN-VQE is unable to find the ground state with comparable accuracy to the RL-based methods, and in some cases fails to solve the problem altogether. This solidifies the fact that having a high-quality initialization alone is insufficient; the RL framework provides a substantial advantage by better guiding the circuit construction and parameter optimization. We will add these insights in the **Results** section in the revised version of the article.
>
> ### **The authors have demonstrated some results about noisy cases, while it would be useful to provide some experimental results on real quantum device to demonstrate its practical use.**
>
> Thank you for this valuable suggestion. We agree that experimental validation on real quantum hardware is important for demonstrating practical relevance. Due to current limitations, primarily long queue times for available QPUs, we focused our main studies on simulations with realistic noise models, including tests with noise levels elevated by 5 and 10 times to probe the robustness of TensorRL-QAS. As shown in [Patel, Yash J., et al., "Curriculum reinforcement learning for quantum architecture search under hardware errors," ICLR (2024)], simulated noise models generally align closely with results from real QPUs. To further address your point, we also evaluated randomly chosen successful (which passes the chemical accurayc) RL circuits generated from noisy simulation with depolarizing noise for 8-$H_2O$ and then run the circuit on $\texttt{IBMQ-Brisbane}$ with $\texttt{Qiskit}$; the results are displayed below which further supporting the practical applicability of TensorRL-QAS.
>
> | Method                         | Noiseless error         | Noisy error (IBMQ-Brisbane)          |
> |---------------------------------|--------------------------|---------------------------|
> | TensorRL (fixed TN-init) (best episode)        | $9.0\times10^{-4}$ |    $2.8\times10^{-3}$   |
> | TensorRL (fixed TN-init)  (random 1)      | $3.4\times10^{-4}$ |    $\mathbf{1.5\times10^{-3}}$   |
> | TensorRL (fixed TN-init)  (random 2)     | $1.2\times10^{-3}$ |    $3.3\times10^{-3}$   |
>
> These results solidifies one of the claim in [Patel, Yash J., et al., "Curriculum reinforcement learning for quantum architecture search under hardware errors," ICLR (2024)] that the best performing circuit in simulated scenario might not be the best performing in QPU.
>
> ### **...the authors may consider to include the results of a pure TN-compiled circuit without RL refinement.**
>
> Thank you for your suggestion regarding a more detailed ablation study. In response, we provide a table of ground-state energies using only pure TN-compiled circuits without RL refinement (TN-energy) for several molecular systems. The error from the reference (true energy) energies serves as a baseline for our method, isolating the contribution of TN compilation alone.
> | Problem | TN-energy | True energy | Error |
> |--------|--------------------|--------------------------|-----------------------|
> | 6-BeH₂    | -14.8556       | -14.8616      | 0.00596              |
> | 8-H₂O          | -73.2914       | -73.2941       | 0.0027              |
> | 10-H₂O         | -74.5633       | -74.5681       | 0.00479              |
> | 12-LiH         | -7.6808        | -7.8746       | 0.19378              |
>
>
> ### **It would be useful to broaden the baseline pool: add recent evolutionary-search and gradient-based QAS methods.**
>
> Thank you for the valuable suggestion. In the revised version of the manuscript we added a gradient-based QAS (GQAS) [He, Zhimin, et al. Neural Networks 179 (2024): 106508.] and DQAS [Zhang, S. X., Hsieh, C. Y., Zhang, S., & Yao, H. (2022). Quantum Science and Technology, 7(4), 045023.] in the table below. This will be added in the **appendix**. Unfortunately for evolutionary-based QAS we could not find any benchmarks that is in line with our optimization method that is why we cite the reference [Zhang, Anqi, and Shengmei Zhao. arXiv preprint arXiv:2212.00421 (2022).] in the **Related works**.
>
> | Problem      | Method                                  | Error              | Depth | CNOT | GATE |
> |--------------|-----------------------------------------|--------------------|-------|------|------|
> | 5-Heisenberg | TensorRL (fixed TN-init)                | $7.6 \times 10^{-2}$   | 15    | 5    | **33**   |
> |  | TensorRL (trainable TN-init, $\chi=2$)  | $3.7 \times 10^{-4}$   | 46    | 28   | 127  |
> |  | StructureRL                             | $2.6 \times 10^{-3}$   | 48    | 33   | 127  |
> |  | TF-QAS                                  | $1.2 \times 10^{-3}$   | NA    | NA   | 35   |
> |  | DQAS (best)                             | $1.11 \times 10^{-1}$  | NA    | NA   | 35   |
> |  | GQAS (best)                             | $7.06 \times 10^{-4}$  | NA    | NA   | 35   |
> |  | GQAS-SSL (best)                         | $\mathbf{4 \times 10^{-6}}$     | NA    | NA   | 35   |
> where **NA** denotes results are not available.
>
> We hope that our response states all the concerns and questions raised by you. Kindly let us know if you have any more suggestions and questions.

---

> > ### Comment · Reviewer_D2Gn · 2025-08-04
> > **Thanks for additional results**
> >
> > I would like to thank the additional results presented by the authors. Some results should be able to be integrated into a revised version of this paper, which may strengthen the quality of this paper.

---

> ### Author Response · Authors · 2025-08-04
>
> Thank you for helping us strengthening our work. We are happy that you appreciated our response. We will include these changes in the camera ready version. Kindly let us know if you have any further feedback.

---

### Official Review · Reviewer_XvsW · 2025-07-02

**Clarity:** 3
**Significance:** 3
**Originality:** 3
**Rating:** 4
**Confidence:** 3

**Summary:**

This paper presents TensorRL-QAS, a framework that makes finding optimal quantum circuits more efficient and scalable with reinforcement learning methods. The core idea is to combine classical tensor networks with RL. First, an approximate solution is found using DMRG and converted into an initial quantum circuit. An RL agent then takes this "warm-start" circuit and intelligently adds more gates to improve it. By starting with a good initial guess, the search for the best circuit is significantly sped up. The paper shows that this method creates much more compact and efficient circuits (with up to 10x fewer CNOT gates) and drastically reduces computation time (by up to 100x) compared to previous approaches. The framework proves effective and robust in both noiseless and noisy simulations, making it a promising tool for near-term quantum computing.

**Questions:**

Is it possible to cooperate the proposed method with real hardware noise model?
How would the "fixed TN-init" model, which relies on preparing an initial statevector, be implemented in practice on current hardware? This step seems non-trivial and may introduce significant overhead not accounted for in the simulations.
How does the performance of TensorRL-QAS change if a different TN-to-circuit mapping is used, or if a different tensor network structure is used for the warm-start?

**Ethical Concerns:**

["NO or VERY MINOR ethics concerns only"]

**Final Justification:**

I generally interested in the paper from the start, and authors mostly addressed my comment, also the comment from Reviewer e9Vh about dmrg also seems addressed by authors. Thus, I keep stand on positive score.

**Limitations:**

see above

**Quality:**

3

**Strengths And Weaknesses:**

The proposed method from this paper is novel and show significant improvement by doing rigorous comprehensive experiments comparing to multiple RL-based and non RL-based methods. Also, this paper provide codebase for reproducibility.
One of the limitation of this paper is that their experiments are based on simulations rather than execution on actual quantum hardware. Also, it would be better to state that whether the choice of different initial structure of MPS could affect the final outcome.

---

> ### Author Rebuttal · Authors · 2025-07-28
>
> We sincerely thank the reviewers for their thoughtful and encouraging feedback. We greatly appreciate your recognition of the novelty of our proposed method, the comprehensive and rigorous nature of our experimental comparisons across RL-based and non-RL-based baselines, and our efforts in providing a reproducible codebase for the community. In the following we address your questions and suggestions.
>
> ### **Is it possible to cooperate the proposed method with real hardware noise model?**
>
> Thank you for this valuable suggestion. We fully agree that validating the proposed method on real hardware is crucial for demonstrating its practical applicability. Due to current limitations, most notably the long queue times associated with available QPUs, our primary studies focus on simulations using realistic noise models. To ensure robustness, we also evaluated the method under elevated noise levels (5x and 10x the baseline) to mimic harsher hardware conditions, as discussed in [Patel, Yash J., et al., "Curriculum reinforcement learning for quantum architecture search under hardware errors," ICLR (2024)]. As shown in that work, simulated noise models often yield trends consistent with actual QPU runs.
>
> To further respond to your point, we conducted an additional experiment: we selected several randomly chosen successful RL-generated circuits (i.e., those achieving chemical accuracy) for 8-$H_2O$ found in noisy simulations (using depolarizing noise), and ran these circuits on $\texttt{IBMQ-Brisbane}$ using $\texttt{Qiskit}$. The results are shown in the table below:
>
> | Method                                      | Noiseless error              | Noisy error (IBMQ-Brisbane)      |
> |--------------------------------------------|-----------------------------|-----------------------------------|
> | TensorRL (fixed TN-init) (best episode)    | $9.0\times10^{-4}$           | $2.8\times10^{-3}$               |
> | TensorRL (fixed TN-init)  (random 1)       | $3.4\times10^{-4}$           | $\mathbf{1.5\times10^{-3}}$      |
> | TensorRL (fixed TN-init)  (random 2)       | $1.2\times10^{-3}$           | $3.3\times10^{-3}$               |
>
> These findings further demonstrate the practical relevance of TensorRL-QAS and support the observation from [Patel et al., ICLR 2024] that the best-performing circuit in simulation may not always be the best on real quantum hardware. This emphasizes the importance of evaluating candidate architectures directly on QPUs when possible and motivates our future work toward a tighter integration with specific hardware noise profiles. We will include this insights and results in the **appendix** of the revised manuscript.
>
> ### **How would the "fixed TN-init" model, which relies on preparing an initial statevector, be implemented in practice on current hardware? This step seems non-trivial and may introduce significant overhead not accounted for in the simulations.**
>
> Thank you for raising this important point regarding the practical implementation of the "fixed TN-init" model, which relies on preparing an initial statevector.
>
> 1. **Compiling TN-init to QPU:** In practice, we would compile the TN-init (tensor network-initialized) state into a quantum circuit using gates natively supported by the target hardware. This allows us to adapt the initialization procedure to the device-specific gateset while ensuring physical realizability.
>
> 2. **Maintaining TN-init on QPU:** If the compiled circuit for the TN-init differs in fidelity from the ideal statevector because of noise, we can employ error mitigation strategies, such as tensor error mitigation [Filippov, Sergei, et al. arXiv preprint arXiv:2307.11740 (2023).], to bring the prepared state as close as possible to the ideal TN-init.
>
> 3. **TensorRL atop the TN-init:** With the TN-init physically implemented as a compiled circuit and error mitigation in place to counteract hardware noise, we can then apply the TensorRL using this fixed initialization. The trained policy and subsequent circuit evolutions can thus be performed atop a stable, experimentally feasible TN-init, while continuing to apply mitigation strategies to maintain fidelity throughout the computation.
>
> Overall, this approach is practically feasible, clearly at the cost of using error mitigation techniques —coming with their unavoidable overheads— to counteract hardware noise. *Due to current limitations in accessing QPUs and limited time on GPUs from our side, a thorough exploration and experimental demonstration of this workflow is left as future work*.
>
> ### **How does the performance of TensorRL-QAS change if a different TN-to-circuit mapping is used, or if a different tensor network structure is used for the warm-start?**
>
> Thank you for raising this important question. Due to time constraints, we were not able to comprehensively investigate alternative TN-to-circuit mappings or entirely different tensor network (TN) structures. However, we sought to probe related effects by varying the number of TN layers in our initialization (from 1 to 3 layers), which closely emulates changes in TN structure and, implicitly, circuit entanglement patterns. For each configuration, we measured the success rate, as well as circuit and error statistics for 6-, 8-, and 10-qubit molecules. The proposed RL agent circuit performance and parameters are summarized in the table below:
>
> | Qubit count | TN layer | Success rate | CNOT | ROT | Depth | Error     |
> |-------------|----------|--------------|------|-----|-------|-----------|
> | 6           | 1        |  0.8478      | 5    | 12  | 7     | **0.00006**   |
> |             | 2        |  0.8418      | 3    | 11  | 5     | 0.00012   |
> |             | 3        |  0.8486      | **2**| **4**| **4**| 0.00024   |
> | 8           | 1        |  0.1423      | 9    | 15  | **6** | **0.00089**   |
> |             | 2        |  0.1398      | **8**| **11**| 9   | 0.00133   |
> |             | 3        |  0.0000      | 6    | 14  | 7     | 0.00196   |
> | 10          | 1        |  0.1288      | 15   | 17  | 17    | **0.00041**   |
> |             | 2        |  0.1097      | **13** | **8** | **9** | 0.00058   |
> |             | 3        |  0           | 18   | 12  | 16    | 0.00178   |
>
> For 6 qubits, both 2- and 3-layer TN initializations perform well, achieving comparably high success rates and competitive circuit costs. However, for 8 and 10 qubits, increasing the TN initialization depth to 3 layers leads to RL failures (resulting in zero successful episodes), whereas the 2-layer initialization maintains reasonable performance. This indicates that deeper or more complex initial circuits, even within the same TN family, can hinder, rather than artificially boost, the overall algorithmic performance. Thus, our findings suggest our method’s success is not attributable solely to an over-optimized or "overfitted" warm-start structure. Instead, the chosen TN initialization achieves a practical trade-off between expressivity and learnability.
>
> While a systematic exploration of alternative TN-to-circuit mappings and fundamentally different TN structures remains a valuable direction for future work, our present experiments with varied TN layers serve as an initial, closely related probe. We will include these additional results and analyses in the **appendix** of the revised manuscript.
>
> We hope our response properly addresses all your questions and suggestions. Kindly let us know if you have any more questions and/or comments.

---

> ### Author Response · Authors · 2025-08-05
>
> Dear reviewer,
>
> We thank you again for your valuable feedback and suggestions on our submission and for raising important points. Below we have carefully addressed each of your comments with detailed clarifications and additional analysis. We kindly invite you to review our responses and share your thoughts, especially regarding the practicality of our proposed approaches on current quantum hardware and their possible overheads. Your further evaluation and guidance will be greatly beneficial to enhance both the work and its presentation.
>
> We look forward to your feedback and a fruitful discussion.
>
> Warm regards,

---

> > ### Comment · Reviewer_XvsW · 2025-08-05
> >
> > I would like to thank authors for the revised version and the detailed explanation. I would like to keep my positive score at the current stage, also, I will track authors rebuttal to reviewer e9Vh - 'how the results would be an improvement beyond the current limits of DMRG' this is a very important question and I am also interested in.

---

> > > ### Author Response · Authors · 2025-08-05
> > >
> > > We sincerely thank you for your feedback, and your engagement with the revision process. We appreciate your interest in the critical question raised by reviewer e9Vh. In the following we summarize the main outcome of response to reviewer e9Vh.
> > >
> > > Our results regarding 'how the results would be an improvement beyond the current limits of DMRG' indicate that for the modest system sizes and accuracies accessible to DMRG on classical hardware, increasing the DMRG bond dimension remains the more straightforward and efficient approach. Our RL-based method does not currently outperform DMRG in this regime. The potential advantage of RL, as we have articulated, would emerge primarily in scenarios where DMRG becomes infeasible, such as for larger, highly entangled systems or in future settings using quantum hardware. However, we acknowledge that realizing such an advantage in practice will require significant further developments, both in terms of algorithmic improvement and quantum resource analysis.
> > >
> > > Kindly check the response to reviewer e9Vh for further clarification and thank you for the positive score.

---

### Note · Authors · 2025-08-11

We sincerely thank the Reviewers and ACs for their service. We are grateful to the reviewers for constructive engagement, it helped refine our claim, strengthened empirical evidence, and improved the manuscript’s clarity. We have worked diligently to address reviewers points and enhance the manuscript in line with the their feedback.

# Strengths
1. Combining DMRG/MPS tensor network initialization with RL for QAS was recognized as novel and promising (D2Gn, XvsW, AdE9).
2. TensorRL-QAS achieves up to 10× CNOT reduction and 100× computation time reduction (D2Gn, XvsW).
3. Acknowledgement of robust performance in noiseless and noisy settings and the outperforms RL and non-RL baselines (D2Gn, XvsW).
5. Presents a reproducible workflow with transparent hyperparameters and public code (D2Gn, XvsW).

# Weaknesses & Responses
1. **Experimental scope (D2Gn, XvsW):** Most results use classical simulation with limited noisy validation.
   - *Response:* We added noisy-simulated circuits executed on IBM quantum hardware; further details in Section 4.2 (camera ready).
2. **Scalability (D2Gn, e9Vh):** Performance at 50+ qubits remains open.
   - *Response:* We enhance RL-based QAS up to 20 qubits and to correctly represent our methodological claim we  replace “scalable” to “enhanced” in the title.
3. **TN-init dependence (AdE9, XvsW, e9Vh):** Impact of TN/Circuit structure and bond dimension.
   - *Response:* In appendix (camera ready) we analyse how bond dimension/TN layers affect RL agent performance.
4. **Ablations & baselines (D2Gn, AdE9):** Request for gradient-based/evolutionary baselines and RL contribution.
   - *Response:* Ablations and new baselines: TN-VQE [1], gradient-based QAS [2,3], evolutionary-based QAS [4]  we add them in camera ready version.
5. **Clarity (e9Vh):** Typos in text and equations.
   - *Response:* Writing and equation typos resolved (camera ready).

**Additional limitation (e9Vh):** In classically tractable regime higher DMRG bond dimension is more efficient than RL. TensorRL-QAS’s future impact is anticipated in NISQ and early fault-tolerant regimes where TN preparation alone is insufficient. Sampling overhead and full hardware-aware RL remain open research challenges.

[1] TN-VQE: DMRG-initialized circuit optimized classically (e.g., COBYLA) without RL.
[2] He, Zhimin, et al. Neural Networks 179 (2024): 106508.
[3] Zhang, S.X., et al. Quantum Sci. Technol. 7(4), 045023 (2022).
[4] Zhang, Anqi, Zhao, S. arXiv:2212.00421 (2022).

---

### Decision · Program_Chairs · 2025-09-17

**Decision:**

Accept (poster)

**Comment:**

This paper studied the problem of ground state preparation using quantum architecture search framework. It utilizes tensor networks especially the DMRG method to prepare an initial ground state, and then applies reinforcement learning to further improve the solution. This framework reduces function evaluations by up to 100-fold, accelerates training episodes by up to 98%, and achieves up to 50% success probability for 10-qubit systems.

Reviewers found this paper with the following strengths:
- The idea of combining DMRG/MPS tensor network initialization with RL for QAS is novel.
- The results of achieving up to 10× CNOT reduction and 100× computation time reduction are competitive.
- The framework has robust performance in noiseless and noisy settings and the outperforms RL and non-RL baselines.

Nevertheless, in the submitted version the reviewers also raised concerns, including scalability, experimental scope, impact of bond dimension, ablations & baselines, etc. The authors addressed these concerns adequately during the rebuttal, and all reviewers reached consensus in accepting this paper. The AC followed this decision accordingly.

Nevertheless, for the final version of the paper, the authors should make the following changes:
- Implement all the promised changes mentioned during rebuttal.
- Include all tables on new experiments conducted during rebuttal.
- When the AC checked the paper, two further points are observed and would be good if the authors can improve: 1) The main body of the paper keeps discussing RL-based algorithms, but didn't introduce much on which RL algorithm was used. It was only vaguely mentioned in Fig. 1 that DDQN was applied. For the NeurIPS audience, it would be helpful to introduce in the main body which RL algorithm was used with which parameter settings, and maybe at least in appendices a comparison between different RL algorithms and why DDQN was the most appropriate choice. 2) There's an important paper on quantum architecture search, the QuantumNAS paper https://ieeexplore.ieee.org/abstract/document/9773233, that was not cited. The authors should discuss about this one as well as some of its follow-up works in quantum architecture literature.